# Batched Thompson Sampling

**Cem Kalkanlı and Ayfer Özgür**
Department of Electrical Engineering
Stanford University
{cemk, aozgur}@stanford.edu

## Abstract

We introduce a novel anytime *batched* Thompson sampling policy for multi-armed bandits where the agent observes the rewards of her actions and adjusts her policy only at the end of a small number of batches. We show that this policy simultaneously achieves a problem dependent regret of order $O(\log(T))$ and a minimax regret of order $O(\sqrt{T \log(T)})$ while the number of batches can be bounded by $O(\log(T))$ independent of the problem instance over a time horizon $T$. We also prove that in expectation the instance dependent batch complexity of our policy is of order $O(\log \log(T))$. These results indicate that Thompson sampling performs competitively with recently proposed algorithms for the batched setting, which optimize the batch structure for a given time horizon $T$ and prioritize exploration in the beginning of the experiment to eliminate suboptimal actions. Unlike these algorithms, the batched Thompson sampling algorithm we propose is an anytime policy, i.e. it operates without the knowledge of the time horizon $T$, and as such it is the only anytime algorithm that achieves optimal regret with $O(\log \log(T))$ expected batch complexity. This is achieved through a dynamic batching strategy, which uses the agents estimates to adaptively increase the batch duration.

## 1 Introduction

The multi-armed bandit problem models the scenario where an agent plays repeated actions and observes rewards associated with her actions. The agent aims to accumulate as much reward as possible, and consequently, she has to balance between playing arms that generated high rewards in the past, i.e. exploitation, and selecting under-explored arms that could potentially return better rewards, i.e. exploration. In the ideal scenario, the agent can adjust her policy once she receives feedback, e.g a reward, before the next action instance. However, many real-world applications limit the number of times the agent can interact with the system. For example, in medical applications [26], many patients are treated simultaneously in each trial, and the experimenter has to wait for the outcome of one trial before planing the next. In online marketing [25], there may be millions of responses per second, and as a result, it is not feasible for the advertiser to update her algorithm every time she receives feedback.

Recently, Perchet et al. [19] proposed to model this problem as the batched multi-armed bandits. Here, the experiment of duration $T$ is split into a small number of batches and the agent does not receive any feedback regarding the rewards of its actions until the end of each batch. For the two-armed bandit problem, they proposed a general class of batched algorithms called explore-then-commit (ETC) policies, where the agent plays both arms the same number of times until the terminal batch and commits to the better performing arm in the last round unless the sample mean of one arm sufficiently dominates the other in earlier batches. They show that this algorithm achieves the optimal problem-dependent regret $O(\log(T))$ and the optimal minimax regret $O(\sqrt{T})$, matching the performance in the classical case where the agent receives instantaneous feedback about her actions, by using only $O(\log(T/\log(T)))$ and $O(\log \log(T))$ batches respectively. Their algorithm takes the time

35th Conference on Neural Information Processing Systems (NeurIPS 2021).

horizon $T$ and divides it into a fixed number of batches before the experiment where the specific batch structure is tuned to the target performance criteria, i.e. problem-dependent or minimax regret. Gao et al. [10] later generalized this result to the setting where the agent had more than two arms to choose from and she could adaptively adjust the batch sizes based on the past data. Their algorithm, called BaSE, is similar to the ETC algorithm in that in each batch the agent plays each of the remaining actions in a round robin fashion, and eliminates the underperforming arms at the end of each batch. They showed that this algorithm required $O(\log(T))$ and $O(\log\log(T))$ number of batches to achieve the optimal problem-dependent and the optimal minimax regret respectively, with batching strategies that were again tailored to the specific objective. More recently several other batched algorithms appeared in the context of asymptotic optimality [13], stochastic and adversarial bandits [9], and linear contextual bandits [11, 22, 20, 21], where the authors provided optimal algorithms in their respective settings. We note that there are also some earlier algorithms developed in the context of the classical bandit setting or bandits with switching cost that even-though not specially developed for the batched setting can be applied in a batched fashion [4, 6].

In this paper, we aim to study whether Thompson sampling, an algorithm that has been developed in 1933 [26] and since then successfully applied to a broad range of online optimization problems [7, 25], can achieve a similar performance in the batched setting. In the Thompson sampling algorithm, the agent chooses an action randomly according to its likelihood of being optimal, and after receiving feedback, i.e. observing rewards, updates its beliefs about the optimal action. The performance of Thompson sampling has been thoroughly analyzed in the literature [16, 17, 1, 2, 23, 24] and is known to achieve the optimal problem-dependent and minimax regret in the classical case. Our goal is to understand whether Thompson sampling can be combined with an adaptive batching strategy and maintain its regret performance when allowed to update its beliefs only at the end of a small number of batches. Note that the earlier algorithms developed specifically for the batched setting [19, 10] heavily prioritize exploration in the initial batches to eliminate the possibility that a suboptimal arm is played in the final exponentially larger batches, while Thompson sampling inherently balances between exploration and exploitation by randomly sampling actions according to their probability of being optimal. A first result in this direction has appeared in [14], which presents a batched Thompson sampling strategy called iPASE. However, [14] provides almost sure guarantees on the asymptotic regret which do not imply the guarantees on expected finite-time regret of interest in this paper and the previously mentioned literature.

Our main contribution in this paper is to show that Thompson sampling, combined with a novel adaptive batching scheme, achieves the optimal problem-dependent performance $O(\log(T))$ and at the same time a minimax regret of order $O(\sqrt{T\log(T)})$ by using $O(\log(T))$ batches independent of the problem instance. This performance is achieved simultaneously by a single strategy without the need to tune the batching structure according to the target criteria, i.e. problem-dependent or minimax regret. Moreover, unlike most of the previously mentioned batched algorithms where $T$ is used both in action selection and the optimization of the batch structure, our strategy is an anytime strategy, i.e. it operates without the knowledge of the time horizon $T$. We note that policies designed for minimizing the problem-dependent regret can indeed be turned into anytime algorithms while retaining their $O(\log(T))$ regret and $O(\log(T))$ batch complexities with the help of the so called doubling trick in [5], but the same extension does not hold for minimax policies. This is because even with the best known doubling schemes (exponential or geometric), the minimax policies either suffer a regret significantly larger than $\sqrt{T}$ or have their batch complexity increase to $\Omega(\log(T))$ (exponential doubling leads to the first conclusion and geometric doubling leads to the second). This implies that our anytime Thompson sampling strategy matches the batch size of $O(\log(T))$ needed for these anytime extensions. In addition, we develop a problem-dependent bound on the expected number of batches used by our strategy which is $O(\log\log(T))$. This shows that while our strategy uses $O(\log(T))$ batches in the worst case, similar to previous algorithms, in a given instance of the problem with fixed reward distributions we only need $O(\log\log(T))$ batches on average. To the best of our knowledge, our scheme is the only strategy that can reduce the number of batches to $O(\log\log(T))$ without the knowledge of the time horizon $T$. This corresponds to a doubly exponential reduction in the interaction needed with the environment as compared to the classical case. We achieve this with a two-fold adaptive batching scheme that allows the batch sizes to increase doubly exponentially as the agent becomes more confident in its decisions. The scheme consists of two layers comprising of batches and multiple cycles inside each batch: the number of cycles per batch is increased exponentially, while at the same time each cycle becomes longer as the agent learns the environment and a suboptimal action is played less and less frequently. This notion of cycles is

key to allowing the batching scheme to dynamically adopt to each instance of the problem, which enables the $O(\log\log(T))$ guarantee on its expected problem-dependent batch complexity.

Finally, we would like to mention a concurrent and independent work by Karbasi et al. [15], also accepted to NeurIPS 2021. They consider the same problem and also develop a batch Thompson sampling strategy for the multi-armed bandits setting. While their results have some overlap with ours, there are also some differences. For example, they establish a $O(\log(T))$ worst-case guarantee on the batch complexity of their algorithm as we do in our paper, however their algorithm lacks a $O(\log\log(T))$ guarantee on its expected batch complexity. In contrast, they generalize their batch Thompson sampling strategy to the linear contextual setting, while we restrict our attention to multi-armed bandits.

## 2  Problem Formulation

### 2.1  Notations

We denote the natural logarithm as $\log(\cdot)$ while a logarithmic function of base $a > 1$ is $\log_a(\cdot)$. For the non-negative sequences of $\{a_n\}$ and $\{b_n\}$, $a_n = O(b_n)$ if and only if $\limsup_{n\to\infty} \frac{a_n}{b_n} < \infty$ and $a_n = \Omega(b_n)$ if and only if $b_n = O(a_n)$. We also denote by $Q(\cdot)$ the probability of a standard normal random variable $X$ being bigger than a certain threshold $x$, i.e. $Q(x) = \mathbb{P}(X \geq x)$ for any $x \in \mathbb{R}$. Finally $\mathbb{1}(\cdot)$ is defined as the indicator function.

### 2.2  Batched Multi-Armed Bandit

We consider the batched multi-armed bandit setting. Here there are $K$ arms, where each consecutive pull of the $i^{th}$ arm produces bounded i.i.d. rewards $\{Y_{i,t}\}_{t=1}^{\infty}$ such that

$$Y_{i,1} \in [0,1],$$
$$\mathbb{E}[Y_{i,1}] = \mu_i \in \mathbb{R}.$$

These mean rewards $\{\mu_i\}_{i=1}^{K}$ are assumed to be deterministic parameters unknown to the agent, whose goal is to accumulate as much reward as possible by repeatedly pulling these arms. Therefore, at each time instance $t$, the agent plays an arm $A_t \in \{1, 2, ..., K\}$ and receives the reward $Y_{A_t, t}$. Since she can only act causally and does not know $\{\mu_i\}_{i=1}^{K}$, she can only use the past observations, $\mathcal{H}_t = \{A_1, Y_{A_t, 1}, ..., A_t, Y_{A_t, t}\}$ where $\mathcal{H}_0 = \emptyset$, to select the next action $A_{t+1}$.

In this paper, we study the batched version of this multi-armed bandit problem, where the agent has to play these arms in batches and can only incorporate the feedback from the system, i.e. her rewards, into her algorithm at the end of a batch. In other words, there are batch end points $0 = T_0 < T_1 < ...$, and the actions the agent plays in the $j^{th}$ batch $[T_{j-1} + 1, T_j]$ as well as the size of the batch itself, i.e. $T_j - T_{j-1}$, can depend only on the information present in $\mathcal{H}_{T_{j-1}}$ for any $j \in \mathbb{Z}^+$ and some external randomness that is independent of the system. Note that in this setting, the agent is allowed to adaptively choose the batch sizes depending on the past observations.

Finally, we let $\mu_1 > \mu_i$ for any $i \geq 2$. Given that the agent aims to maximize her cumulative reward, she would only play the first arm if she knew the hidden system parameters $\{\mu_i\}_{i=1}^{K}$. This observation naturally leads to the cumulative regret term, $R(T)$:

$$\mathbb{E}[R(T)] = \sum_{i=2}^{K} \Delta_i \, \mathbb{E}[N_i(T)] \tag{1}$$

where $\mu_1 - \mu_i = \Delta_i$ and

$$N_i(T) = \sum_{t=1}^{T} \mathbb{1}(A_t = i)$$

for $i \in \{1, 2, ..., K\}$.

# 3 Batched Thompson Sampling

In this section, we describe our Batched Thompson sampling strategy for the batched multi-armed bandit setting described in the previous section. This policy uses Gaussian priors in the spirit of [2] and each arm is sampled randomly according to its likelihood of being optimal under this prior and the observations from previous batches. We combine this strategy with an batching mechanism which relies on the notion of *cycles*. A cycle is defined as follows. The first cycle starts in the beginning of the experiment and ends when the agent selects an action different from the previous actions, i.e. it corresponds to the shortest time interval starting from the beginning of the experiment where two different actions are selected. Then the $j^{th}$ cycle for $j > 1$ is defined recursively as the time interval from the end of the $j - 1^{th}$ cycle to the first time step where the agent selects an action different from the first action in the cycle. In other words, in each cycle the agent plays exactly two different actions. Consider the following example. Assume that the first seven actions played by the agent are as follows: $A_1 = 1$, $A_2 = 1$, $A_3 = 2$, $A_4 = 1$, $A_5 = 3$, $A_6 = 2$, $A_7 = 2$. Then the first cycle is $[1, 3]$ because only at the third time step the agent selected an arm different from the earlier actions. Similarly, the second cycle is $[4, 5]$ where the agent played the first and the third actions. The third cycle that started on $t = 6$ has not ended yet because only a single action has been played so far. We use the concept of a cycle to adaptively decide on the batch size. At the beginning of the $j^{th}$ batch, the Thompson sampling agent checks the number of cycles in which each action $i$ has been played since the beginning of the experiment, denoted $M_i(T_{j-1})$, and sets upper limits $U_{i,j} = \max\{1, \lceil \alpha \times M_i(T_{j-1}) \rceil\}$ for the cycle count of each action. Here $\alpha > 1$ is a batch growth factor to be chosen. Throughout the $j^{th}$ batch, the agent employs Thompson sampling, and at the end of each cycle checks whether or not the number of cycles in which a certain arm has been selected since the beginning of the experiment has reached its upper limit $U_{i,j}$ set for the current batch. The batch ends if there is one such action hitting its upper limit. After the $j^{th}$ batch, the agent observes the rewards of its actions and repeats the same process in the next batch. Note that as the algorithm proceeds the batch size increases due to two different reasons. First, the cycle count of each action is increased at the end of each batch. This alone leads to an exponential increase in the number of cycles per batch with base $\alpha$. However, in addition to this, the length of each cycle becomes longer as the agent becomes more confident about its decisions and the probability of playing an optimal arm decreases. Note that the exponential increase in the number of cycles per batch is there even if the algorithm does not make progress in learning its environment, e.g. always plays arms with equal or near-equal probability due to a very noisy environment. However, the increase in the cycle duration is tightly coupled with the confidence of the agent and the cycle duration increases only if the agent becomes more confident. Therefore, this notion of cycles allows us to adopt the batch size much more dynamically to the current progress of the agent in a given instance of the problem.

We now introduce the following notation to denote the beginning and end of the $k^{th}$ cycle, $C_{b,k}$ and $C_{e,k}$ respectively:

$$C_{b,k} = \begin{cases} 1 & \text{if } k = 1 \\ C_{e,k-1} + 1 & \text{if } k > 1 \end{cases}$$

and

$$C_{e,k} = \min\{t \in \mathbb{Z}^+ | A_t \neq A_{t-1} \text{ and } t > C_{b,k}\}$$

for any positive integer $k$. As can be seen from these definitions, the interval $[C_{b,k}, C_{e,k}]$ describes the $k^{th}$ cycle. We also define $M_i(T)$ and $S_i(T)$ as follows

$$M_i(T) = \sum_{t=1}^{T} \mathbb{1}(A_t = i, t = C_{b,k} \text{ or } C_{e,k} \text{ for some } k)$$

and

$$S_i(T) = \sum_{t=1}^{T} \mathbb{1}(A_t = i, t = C_{b,k} \text{ or } C_{e,k} \text{ for some } k) Y_{i,t}.$$

Here $M_i(T)$ denotes the number of cycles in which the $i^{th}$ action has been selected, while $S_i(T)$ is the sum of rewards the agent received from playing the $i^{th}$ action at either the beginning or the end of a cycle over the duration $T$ of the experiment. Note that whether the condition $\{t = C_{b,k} \text{ or } C_{e,k} \text{ for some } k\}$ is satisfied or not can be verified by checking the actions taken until the time step $t$, i.e. $\{A_j\}_{j=1}^{t}$. We also define $b(t) = \max\{j \in \mathbb{Z}_{\geq 0} | t - 1 \geq T_j\}$ as the index of the

last batch, and $B(t) = \min\{j \in \mathbb{Z}^+ | t \leq T_j\}$ as the batch index of the $t^{th}$ time step. We provide a pseudo-code for our Batched Thompson Sampling policy in Algorithm 1.

---

**Algorithm 1:** Batched Thompson Sampling

---

**Input:** Batch growth factor $\alpha > 1$, Gaussian variance $\sigma^2$
**Initialization:** $t = 1$, $M_i(0) = 0$, $S_i(0) = 0$, $U_{i,1} = 1$, $j = 1$, $T_0 = 0$.
**while** *Experiment Run* **do**

    **Sample for Each Arm:** $\theta_i(t) \sim \mathcal{N}(\frac{S_i(T_{j-1})}{1+M_i(T_{j-1})}, \frac{\sigma^2}{1+M_i(T_{j-1})})$

    **Play an Arm:** $A_t = \arg\max_i \theta_i(t)$

    **Update the Pull Count:** $M_i(t) \leftarrow M_i(t-1) + \mathbb{1}(A_t = i, t = C_{b,k} \text{ or } C_{e,k} \text{ for some } k)$

    **if** $t = C_{e,k}$ *for some k* **&** $M_i(t) = U_{i,j}$ *for some i* **then**

        **End the Current Batch:** $T_j = t$

        **Receive the Rewards:** $\{Y_{A_t,t}\}_{t=T_{j-1}+1}^{T_j}$

        **Update the Cumulative Rewards for Each Arm:**

        $S_i(T_j) = S_i(T_{j-1}) + \sum_{l=T_{j-1}+1}^{T_j} \mathbb{1}(A_l = i, l = C_{b,k} \text{ or } C_{e,k} \text{ for some } k)Y_{i,l}$

        **Update the Upper Limites for the Cycle Counts:** $U_{i,j+1} = \max\{1, \lceil \alpha \times M_i(T_j) \rceil\}$

        **Update the Batch Index:** $j \leftarrow j + 1$

    **end**

    **Update the Time Index:** $t \leftarrow t + 1$

**end**

---

Note that in Algorithm 1, the posterior distribution from which each arm is selected depends only on $\{M_i(T_{b(t)}), S_i(T_{b(t)})\}_{i=1}^K$, that is we only use the rewards from the first and last action selected in each cycle and ignore the rest of the observed rewards. This is to simplify the analysis in the following sections. However, we can also apply the algorithm by incorporating all the observed rewards, which in general can yield better performance while still maintaining the same batch structure. In that case, $\theta_i(t)$ for any $t$ in the $j^{th}$ batch is drawn instead as

$$\theta_i(t) \sim \mathcal{N}\Big(\frac{\sum_{t=1}^{T_{j-1}} \mathbb{1}(A_t = i)Y_{i,t}}{1 + N_i(T_{j-1})}, \frac{\sigma^2}{1 + N_i(T_{j-1})}\Big).$$

## 4 Main Results

In this section, we state the main results of our paper. We start with the regret performance of Batched Thompson sampling.

**Theorem 1.** *Consider the batched multi-armed bandit setup described in Section 2. If $T \geq 2$ and the batch growth factor $\alpha$ satisfies $1 < \alpha \leq \frac{5\sigma^2}{4}$, then Batched Thompson sampling obeys the following inequalities*

$$\mathbb{E}[N_i(T)] \leq C_1 \alpha \sigma^2 \frac{\log(T)}{\Delta_i^2} \tag{2}$$

*for any $i \geq 2$, which lead to*

$$\mathbb{E}[R(T)] \leq C_1 \alpha \sigma^2 \sum_{i=2}^K \frac{\log(T)}{\Delta_i}, \tag{3}$$

*and*

$$\mathbb{E}[R(T)] \leq C_2 \sigma \sqrt{\alpha K T \log(T)} \tag{4}$$

*where $C_1, C_2 \geq 1$ are absolute constants independent of the system parameters.*

We provide the proof of this theorem for the special case of $K = 2$, $\alpha = 2$, $\sigma^2 = 1$ in Section 6.1, and defer the proof of the general version to the supplementary materials.

Theorem 1 states that Batched Thompson sampling achieves $O(\log(T) \sum_{i=2}^K \Delta_i^{-1})$ problem-dependent regret and $O(\sqrt{KT \log(T)})$ minimax regret, which match the asymptotic lower bound of $\Omega(\log(T))$ [18] and the minimax lower bound of $\Omega(\sqrt{KT})$ [3] up to a $\sqrt{\log(T)}$ term respectively.

We next compare our bounds with the results on classical Thompson sampling [2]. As described in the previous section, we use Gaussian priors for Thompson sampling following the work of Agrawal and Goyal [2]. This is one of the two priors considered in that work for Thompson sampling: beta priors and Gaussian priors. For Thompson sampling with Beta priors, Agrawal and Goyal [2] provides two different bounds on the expected regret:

$$\mathbb{E}[R(T)] \leq (1+\epsilon) \sum_{i=2}^{K} \frac{\log(T)}{d(\mu_i, \mu_1)} \Delta_i + O(\frac{K}{\epsilon^2}), \tag{5}$$

for any $\epsilon \in (0, 1)$, and

$$\mathbb{E}[R(T)] \leq O(\sqrt{KT \log(T)}), \tag{6}$$

where $d(\mu_i, \mu_1) = \mu_i \log(\frac{\mu_i}{\mu_1}) + (1 - \mu_i) \log(\frac{1-\mu_i}{1-\mu_1})$. In addition, they show that with Gaussian priors the expected regret of the classical Thompson sampling is bounded by $O(\sqrt{KT \log(K)})$ if $T \geq K$. Considering that $d(\mu_i, \mu_1) \geq 2\Delta_i^2$ by Pinsker's inequality, (5) provides a tighter performance guarantee than (3) in terms of the dependence on the reward distributions, but we note that the minimax performance of Batched Thompson sampling, (4), matches the performance of classical Thompson sampling when the agent receives instantaneous feedback about rewards and can update its policy after each action. These results show that Batched Thompson sampling, apart from the dependence on the reward distributions in the problem dependent bound, matches the regret performance in the classical case.

We note that the regret bounds in the theorem depend on the batch growth factor $\alpha$. The regret increases linearly as $\alpha$ grows bigger; this is not surprising since bigger batch sizes mean fewer updates for Batched Thompson sampling.

We now present the batch complexity results for our algorithm.

**Theorem 2.** *Consider the batched multi-armed bandit setup described in Section 2. If $T \geq 2$ and $1 < \alpha \leq \frac{5\sigma^2}{4}$, then the batch complexity $B(T)$ of Batched Thompson sampling satisfies the following:*

$$B(T) \leq 1 + K + K \log_\alpha(1 + \frac{T}{K}), \tag{7}$$

$$\mathbb{E}[B(T)] \leq 1 + K + \log_\alpha(1 + C\alpha\sigma^2 \sum_{i=2}^{K} \frac{\log(T)}{\Delta_i^2})) + \sum_{i=2}^{K} \log_\alpha(1 + C\alpha\sigma^2 \frac{\log(T)}{\Delta_i^2}), \text{ and} \tag{8}$$

$$\mathbb{E}[B(T)] \leq 1 + 2C\alpha\sigma^2 \sum_{i=2}^{K} \frac{\log(T)}{\Delta_i^2}, \tag{9}$$

*where $C$ is an absolute constant independent of the system parameters.*

Theorem 2 states three different batch complexity guarantees; the first is a deterministic guarantee on the number of batches while the last two bound the number of batches in expectation. If we consider (7), we see that Batched Thompson sampling uses at most $O(\log(T))$ many batches regardless of the reward distributions. This result and Theorem 1 indicate that Batched Thompson sampling matches the regret performance and the batch complexities of optimal problem-dependent batched algorithms developed in [4, 10, 9, 15], which also achieve $O(K \log(T))$ problem-dependent regret with $O(\log(T))$ batch complexity. However, compared with the other optimal algorithms, we show that we can further reduce the expected batch complexity down to $O(K \log \log(T))$ in (8). This is because our batching strategy uses the information it gathers about the system (through the notion of cycles) to adaptively decide on the sizes of the batches while most prior batched algorithms use a static batch structure. We note that Algorithm 1 of Esfandiari et al. [9] does use an adaptive batching strategy however that strategy appears to be geared towards obtaining a tighter regret bound rather than reducing the batch complexity.

We note that the bounds on the number of batches in (7) and (8) diverge to infinity as $\alpha \downarrow 1$. The bound in (9) on the other hand decreases with $\alpha$ and can be relevant when $\alpha$ is chosen very small. We also note that if $\alpha < 1 + \frac{1}{T}$ for a fixed $T$, then Batched Thompson sampling will only allow one cycle per batch throughout the experiment of duration $T$ and the notion of a batch will coincide with the

notion of a cycle. In this extremal case with one cycle per batch, (9) shows that Batched Thompson sampling will have $O(\log(T))$ batch complexity, while still satisfying the regret bounds stated in Theorem 1. The $O(K \log\log(T))$ bound on the expected batch complexity in (8) is enabled by the fact that for larger $\alpha$ we allow for exponentially more cycles in each batch. As the algorithm proceeds and becomes more confident about the system, choosing a suboptimal arm becomes less likely and as a result the cycle durations become inherently larger. At the same time, the algorithm allows for exponentially more cycles in each batch. This double batching strategy in our algorithm, via cycles and batches is key to obtain $O(K \log\log(T))$ guarantee in (8) with an anytime strategy. Note that the best previously available guarantee on batch complexity for an anytime batching strategy is $O(\log(T))$.

The proof of these theorems are provided in Section 6. The main technical novelty in the proof comes from the fact that the notion of cycles leads to a random duration for each batch without any deterministic upper bound on the number of times each arm is played in the batch. This makes it more difficult to control the regret accumulated during a given batch. We note that without the notion of cycles (e.g. when a cycle corresponds to a pull of an arm) the exponential increase in the number of arms per batch is relatively easy to deal with, as in this case we have a deterministic upper bound on the number of times each arm can be played in a given batch.

## 5  Experiments

In this section, we provide some experimental results on the performance of Batched Thompson sampling where we do not skip samples, i.e. the variant mentioned at the end of Section 3, for different values of $\alpha$, $\{1.00001, 1.25, 1.5, 2\}$, and how they perform against normal Thompson sampling under different reward distributions and action counts when time horizon $T = 5 \times 10^4$ and the sampling variance $\sigma^2 = 1$. We mainly consider four setups: Bernoulli rewards when $K = 2$, Figure 1 (a); Bernoulli rewards when $K = 5$, Figure 1 (b); Gaussian rewards when $K = 2$, Figure 1 (c); Gaussian rewards when $K = 5$, Figure 1 (d). Finally each figure is the result of an experiment averaged over $10^4$ repeats and the average number of batches used throughout the experiment is rounded up to the nearest integer and reported in the parenthesis to the right of the algorithm names on the figures. The figures show that our batching strategy matches the performance of classical Thompson sampling by using roughly 100 batches over a time horizon of $T = 5 \times 10^4$.

As can be seen from Figure 1, Batched Thompson sampling achieves almost the same empirical performance as the normal Thompson sampling when we set $\alpha$ small enough so that there is only one cycle per batch, i.e. $\alpha = 1.00001$. We also observe that this Batched Thompson sampling version $\alpha = 1.00001$, can have a batch count as small as 15. However when $\alpha$ is very small, the problem independent guarantees in (7) and (8) become very loose and the number of batches can vary more with the reward distributions. This can be partly observed in the figures: for $\alpha = 1.00001$, there is larger variation in the average batch complexity across different reward distributions though in all cases the average numbe of batches remain very small. Increasing $\alpha$ leads to a more stable batch complexity behavior, at the cost of a small multiplicative regret factor; we observe that the batch count almost remains constant for $\alpha = 2$ across different reward distributions. The source codes of the experiment can be found in `https://github.com/incsmi/BatchedThompsonSampling.git`.

## 6  Technical Analysis

In this section we provide technical proofs for our results. We start with the proof of Theorem 1 for a special case and at the end prove Theorem 2.

### 6.1  Proof of Theorem 1 when $K = 2$, $\alpha = 2$, and $\sigma^2 = 1$

We first introduce $N_{2,j}(t)$ as the number of times the second arm is pulled if Batched Thompson sampling is employed for $t$ many round with the past knowledge of $\mathcal{H}_{T_{j-1}}$. In this case, $N_{2,j}(T_j - T_{j-1}) = N_2(T_j) - N_2(T_{j-1})$. We know that in the first $T$ rounds, there can be no more than $T$ many batches, and each batch can not last longer than $T$ rounds. As a result, we have the following bound

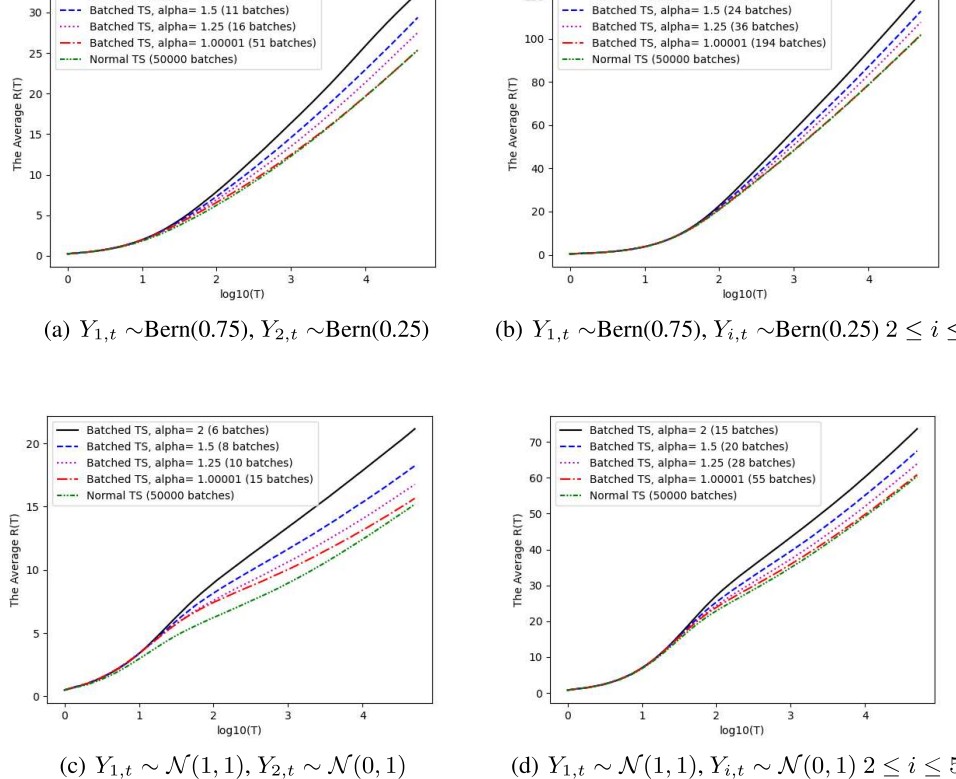

(a) $Y_{1,t} \sim \text{Bern}(0.75)$, $Y_{2,t} \sim \text{Bern}(0.25)$

(b) $Y_{1,t} \sim \text{Bern}(0.75)$, $Y_{i,t} \sim \text{Bern}(0.25)$ $2 \leq i \leq 5$

(c) $Y_{1,t} \sim \mathcal{N}(1,1)$, $Y_{2,t} \sim \mathcal{N}(0,1)$

(d) $Y_{1,t} \sim \mathcal{N}(1,1)$, $Y_{i,t} \sim \mathcal{N}(0,1)$ $2 \leq i \leq 5$

Figure 1: Empirical Regret Performance of Batched and normal Thompson sampling

on $N_2(T)$:

$$N_2(T) \leq \sum_{j=1}^{T} \min(N_{2,j}(T_j - T_{j-1}), N_{2,j}(T)).$$

Now we first analyze the expected number of times the second is pulled in the first cycle of the $j^{th}$ batch. It is easy to see the time the first arm is selected in this cycle is an upper bound on the number of times the second arm is picked. This observation follows from the fact that if the first action is selected in the first round, then the second is selected only once in the current cycle, if not the time the first arm is selected becomes one more than the number of times the second arm is picked. As a result, conditioned on the past $\mathcal{H}_{T_{j-1}}$, the expected number of times the second arm is picked in a single cycle is upper bounded by $\frac{1}{\mathbb{P}(A_{T_{j-1}+1}=1|H_{T_{j-1}})}$. Also note that since there are only two actions, each cycle will contain both of them, and in the first $j$ batches, there will be $2^{j-1}$ many cycles by the construction of Algorithm 1. This observation means that there are no more than $2^{j-1}$ many identically distributed such cycles in the $j^{th}$ batch, and we have the following bound for any $j$:

$$\mathbb{E}[N_{2,j}(T_j - T_{j-1})] \leq 2^{j-1} \, \mathbb{E}\Big[\frac{1}{\mathbb{P}(A_{T_{j-1}+1} = 1|H_{T_{j-1}})}\Big] \leq C2^{j-1} \tag{10}$$

where $C \geq 2$ is a constant independent of $j$. The last inequality in (10) follows from the following Lemma 3 and the fact that in the first batch $\mathbb{P}(A_{T_{j-1}+1} = 1|H_{T_{j-1}}) = \mathbb{P}(A_1 = 1) = \frac{1}{2}$.

**Lemma 3.** *If $K = 2$, $\alpha = 2$, and $\sigma^2 = 1$, then any $j \geq 2$:*

$$\mathbb{E}\left[\frac{1}{\mathbb{P}(A_{T_{j-1}+1} = 1|H_{T_{j-1}})}\right] \leq C, \tag{11}$$

$$\mathbb{P}(A_{T_{j-1}+1} = 2) \leq \exp(-\frac{2^{j-4}}{3}\Delta_2^2), \tag{12}$$

*where $C \geq 2$ is a constant independent of $j$.*

In addition, we also have:

$$\mathbb{E}[N_{2,j}(T)] = T\,\mathbb{P}(A_{T_{j-1}+1} = 2) \leq T\exp(-\frac{2^{j-4}}{3}\Delta_2^2) \tag{13}$$

for $j \geq 2$ by the same lemma. The overall analysis shows that for any positive integer $k$, we have:

$$\mathbb{E}[N_2(T)] \leq \mathbb{E}[\sum_{j=1}^{T}\min(N_{2,j}(T_j - T_{j-1}), N_{2,j}(T))] \leq \sum_{j=1}^{k}\mathbb{E}[N_{2,j}(T_j - T_{j-1})] + \sum_{j=k+1}^{T}\mathbb{E}[N_{2,j}(T)]$$

$$\leq C(2^k - 1) + T\sum_{j=k+1}^{T}\exp(-\frac{2^{j-4}}{3}\Delta_2^2) \tag{14}$$

where the last step follows from (10) and (13). Let $k$ be the smallest positive integer such that $\frac{2^k}{3} \geq 8\frac{\log(T)}{\Delta_2^2}$. Then we have

$$T\sum_{j=k+1}^{T}\exp(-\frac{2^{j-4}}{3}\Delta_2^2) \leq T\sum_{i=0}^{\infty}\exp(-2^i\frac{2^{k-3}}{3}\Delta_2^2) \leq T\sum_{i=0}^{\infty}\exp(-2^i\log(T))$$

$$= \sum_{i=0}^{\infty}\frac{1}{T^{2^i-1}} \leq \sum_{i=0}^{\infty}\frac{1}{T^i} \leq 2$$

where the last inequality follows from the assumption that $T \geq 2$. This analysis bounds the last summation term in (14). To bound $C(2^k - 1)$, note that $k$ is the smallest positive integer bigger than $\log_2(24\frac{\log(T)}{\Delta_2^2})$. Since $\Delta_2^2 \leq 1$ and $T \geq 2$, we know $24\frac{\log(T)}{\Delta_2^2} \geq 1$. This analysis shows that $k \leq \log_2(24\frac{\log(T)}{\Delta_2^2}) + 1 = \log_2(48\frac{\log(T)}{\Delta_2^2})$. As a result, $C(2^k - 1) \leq 48C\frac{\log(T)}{\Delta_2^2}$. The overall analysis shows that $\mathbb{E}[R(T)] = \Delta_2\,\mathbb{E}[N_2(T)] \leq 48C\frac{\log(T)}{\Delta_2} + 2\Delta_2$ by (14). This finishes the proof of (3). Finally (4) is proven by (3) if $\Delta_2 > \sqrt{\frac{\log(T)}{T}}$. If not, then $\mathbb{E}[R(T)] \leq T\Delta_2 \leq \sqrt{T\log(T)}$, and this proves (4).

### 6.2  Proof of Theorem 2

Let us consider the case where the agent has already employed the Batched Thompson sampling, Algorithm 1, for $T$ many steps and denote $i_j \in \{1, 2, ..., B(T) - 1\}$ for $j \in \{1, 2, ..., k_i\}$ as the indices where $M_i(T_{i_j}) = U_{i,i_j}$. Since each batch end point $T_j$ has to satisfy the condition $M_l(T_j) = U_{l,j}$ for some $l$, we have

$$B(T) - 1 \leq \sum_{i=1}^{K}k_i. \tag{15}$$

By the definition of $U_{i,j}$, we know that $M_i(T_{i_1}) = 1$. In addition, note that there may be batches in between $i_{j-1}^{th}$ and $i_j^{th}$ ones and the agent may have picked the $i^{th}$ arm while the condition $M_i(T_j) = U_{i,j}$ is not satisfied. These observations lead to $\max\{\alpha M_i(T_{i_{j-1}}), M_i(T_{i_{j-1}}) + 1\} \leq M_i(T_{i_j})$ for $j \geq 2$. The overall analysis shows that if $k_i \geq 1$, then $\alpha^{k_i-1} \leq M_i(T_{i_{k_i}}) \leq M_i(T_{B(T)-1})$ due to the fact that $T_{i_{k_i}} \leq T_{B(T)-1}$, which leads to $k_i \leq 1 + \log_\alpha(1 + M_i(T_{B(T)-1}))$ for any $k_i \geq 0$. In addition, we also have the following trivial bound $k_i \leq M_i(T_{i_{k_i}}) \leq M_i(T_{B(T)-1})$. As a result of these inequalities and (15), we have

$$B(T) \leq 1 + K + \sum_{i=1}^{K}\log_\alpha(1 + M_i(T_{B(T)-1})) \tag{16}$$

and

$$B(T) \leq 1 + \sum_{i=1}^{K}M_i(T_{B(T)-1}). \tag{17}$$

First of all, since $\log(\cdot)$ is a concave function, Jensen's inequality and (16) lead to

$$B(T) \le 1 + K + K \log_\alpha (1 + \frac{1}{K} \sum_{i=1}^{K} M_i(T_{B(T)-1})). \tag{18}$$

Considering that each cycle has to contain at least two action steps, there can be no more than $\frac{T}{2}$ many cycles in the first $B(T) - 1$ batches. In addition, each cycle can only be recorded once by two different actions. This observation leads to $\sum_{i=1}^{K} M_i(T_{B(T)-1}) \le T$, which proves (7) by (18). To prove (8), we first note that $M_1(T_{B(T)-1}) \le \sum_{i=2}^{K} M_i(T_{B(T)-1})$ because each cycle of the first arm has to be accompanied by another arm. Since $M_i(T_{B(T)-1}) \le N_i(T)$ for any $i$, we have $B(T) \le 1 + K + \log_\alpha (1 + \sum_{i=2}^{K} N_i(T)) + \sum_{i=2}^{K} \log_\alpha (1 + N_i(T))$ by (16), which shows that

$$\mathbb{E}[B(T)] \le \mathbb{E}[1 + K + \log_\alpha (1 + \sum_{i=2}^{K} N_i(T)) + \sum_{i=2}^{K} \log_\alpha (1 + N_i(T))]$$

$$\le 1 + K + \log_\alpha (1 + \sum_{i=2}^{K} \mathbb{E}[N_i(T)]) + \sum_{i=2}^{K} \log_\alpha (1 + \mathbb{E}[N_i(T)]).$$

The last inequality follows from Jensen's inequality. This leads to (8) by the fact that $\mathbb{E}[N_i(T)] \le C\alpha\sigma^2 \frac{\log(T)}{\Delta_i^2}$ from (2). Finally, the previous analysis also implies that $B(T) \le 1 + 2\sum_{i=2}^{K} N_i(T)$ by (17). This inequality and (2) prove (9).

# 7 Conclusion

We proposed an anytime Batched Thompson sampling algorithm and proved that it achieves the optimal problem-dependent and minimax regret with only $O(\log(T))$ instance-independent batch complexity, matching the state-of-the-art anytime batched algorithms. More interestingly, we showed that in a given instance of the problem our algorithm requires only $O(\log \log(T))$ batches on average, which corresponds to a doubly exponential decrease in the interaction needed with the environment as compared to the classical case. To the best of our knowledge, previous anytime algorithms only satisfy a worst-case $O(\log(T))$ guarantee on the batch complexity. Finally, simulations show that Batched Thompson sampling performs empirically close to classical Thompson sampling by using drastically fewer batches.

# 8 Acknowledgements

This work was supported in part by a Google Faculty Research Award, and the National Science Foundation under grants CCF1704624 and NeTS-1817205.

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
