## A  Outline

The appendix is organized as follows.

1. Section B states technical tools necessary for our proofs.
2. Section C provides the proof of Lemma 3, which is stated in Section 6.1.
3. In Section D, we present couple propositions and lemmas in preparation for the proof of Theorem 1 in the general case of $K$, $\alpha$, and $\sigma^2$.
4. The full proof of Theorem 1 is finally given in Section E.
5. In Section F, we provide additional experiments in which we compare different variants of Batched Thompson sampling: the one using all the observations and the one skipping samples of multiple instances from the same arm.

## B  Technical Tools

### B.1  Bounded Random Variable Moment-generating Function Bound

Let $X$ be a bounded zero mean random variable such that $a \leq X \leq b$ and $|a|, |b| < \infty$. Hoeffding [12] showed that

$$\mathbb{E}[\exp(\lambda X)] \leq \exp(\lambda^2 \frac{(b-a)^2}{8}) \tag{19}$$

for any real number $\lambda$.

### B.2  Gaussian Tail Bounds

Proposition 2.1.2 of [27] shows that

$$\left(\frac{1}{\delta} - \frac{1}{\delta^3}\right) \frac{\exp(-\frac{\delta^2}{2})}{\sqrt{2\pi}} \leq Q(\delta) \leq \frac{1}{\delta} \frac{\exp(-\frac{\delta^2}{2})}{\sqrt{2\pi}}, \tag{20}$$

if $\delta > 0$. Since exponential functions decay faster than power functions, there exists $\delta_0$ such that if $\delta \geq \delta_0$, then

$$\exp(-3\delta^2/4) \leq Q(\delta)$$

which leads to

$$Q^{-1}(1/x) \geq \sqrt{\frac{4}{3} \log(x)} \tag{21}$$

if $x \geq x_0$ for some $x_0 \geq 2$, where $Q^{-1}(\cdot)$ is the inverse function of $Q(\cdot)$. Note that the last inequality follows from setting $\delta = \sqrt{\frac{4}{3} \log(x)}$ and the fact that $Q(\cdot)$ is decreasing.

### B.3  Expectation of Non-negative Random Variables

Let $X$ be a non-negative random variable, i.e. $X \geq 0$, then

$$\mathbb{E}[X] = \int_0^\infty \mathbb{P}(X > x) dx \tag{22}$$

by Lemma 2.2.13 of [8].

## C  Proof of Lemma 3

First of all,

$$\mathbb{E}\left[\frac{1}{\mathbb{P}(A_{T_{j-1}+1} = 1|H_{T_{j-1}})}\right] = \int_0^\infty \mathbb{P}\left(\frac{1}{\mathbb{P}(A_{T_{j-1}+1} = 1|H_{T_{j-1}})} > x\right) dx$$

$$\leq 2 + \int_2^\infty \mathbb{P}(\mathbb{P}(A_{T_{j-1}+1} = 1|H_{T_{j-1}}) \leq \frac{1}{x}) dx \tag{23}$$

by (22). Conditioned on $\mathcal{H}_{T_{j-1}}$, $\theta_i(T_{j-1}+1)$ is distributed as $\mathcal{N}(\frac{S_i(T_{j-1})}{1+M_i(T_{j-1})}, \frac{1}{1+M_i(T_{j-1})})$. However when $K = 2$ and $\alpha = 2$, we know that $M_i(T_{j-1}) = 2^{j-2}$. This overall analysis leads to

$$
\begin{aligned}
\mathbb{P}(A_{T_{j-1}+1} = 1 | H_{T_{j-1}}) &= \mathbb{P}\left(\mathcal{N}\left(\frac{S_1(T_{j-1})}{1+2^{j-2}}, \frac{1}{1+2^{j-2}}\right) \geq \mathcal{N}\left(\frac{S_2(T_{j-1})}{1+2^{j-2}}, \frac{1}{1+2^{j-2}}\right) | H_{T_{j-1}}\right) \\
&= \mathbb{P}\left(\mathcal{N}\left(\frac{S_1(T_{j-1}) - S_2(T_{j-1})}{1+2^{j-2}}, \frac{2}{1+2^{j-2}}\right) \geq 0 | H_{T_{j-1}}\right) \\
&= 1 - \mathbb{P}\left(\mathcal{N}\left(0, \frac{2}{1+2^{j-2}}\right) \geq \frac{S_1(T_{j-1}) - S_2(T_{j-1})}{1+2^{j-2}} | H_{T_{j-1}}\right) \\
&= 1 - Q\left(\frac{S_1(T_{j-1}) - S_2(T_{j-1})}{\sqrt{2+2^{j-1}}}\right)
\end{aligned}
$$

where the last equality follows from the definition of the function $Q(\cdot)$. Combining the last analysis with (23) shows that

$$
\begin{aligned}
\mathbb{E}\left[\frac{1}{\mathbb{P}(A_{T_{j-1}+1} = 1 | H_{T_{j-1}})}\right] &\leq 2 + \int_2^\infty \mathbb{P}\left(1 - Q\left(\frac{S_1(T_{j-1}) - S_2(T_{j-1})}{\sqrt{2+2^{j-1}}}\right) \leq \frac{1}{x}\right) dx \\
&= 2 + \int_2^\infty \mathbb{P}\left(Q\left(\frac{S_1(T_{j-1}) - S_2(T_{j-1})}{\sqrt{2+2^{j-1}}}\right) \geq 1 - \frac{1}{x}\right) dx \\
&= 2 + \int_2^\infty \mathbb{P}\left(\frac{S_2(T_{j-1}) - S_1(T_{j-1})}{\sqrt{2+2^{j-1}}} \geq -Q^{-1}\left(1 - \frac{1}{x}\right)\right) dx \quad (24) \\
&= 2 + \int_2^\infty \mathbb{P}\left(\frac{S_2(T_{j-1}) - S_1(T_{j-1})}{\sqrt{2+2^{j-1}}} \geq Q^{-1}\left(\frac{1}{x}\right)\right) dx \quad (25)
\end{aligned}
$$

where (24) follows from the fact that $Q(\cdot)$ is a decreasing function, and (25) is the result of the symmetric nature of the normal distribution.

Let $\lambda$ be any real number. Here we are going to use induction hypothesis. We know that $\mathbb{E}[\exp(\lambda(S_2(T_1) - S_1(T_1) + \Delta_2))] \leq \exp(\lambda^2/4)$ by (19) and the fact that $S_1(T_1)$ and $S_2(T_1)$ are independent bounded random variables. Now assume that $\mathbb{E}[\exp(\lambda(S_2(T_j) - S_1(T_j) + 2^{j-1}\Delta_2))] \leq \exp(2^{j-3}\lambda^2)$ for some $j \geq 1$. However we know that $\mathcal{H}_{T_j}$, $S_1(T_{j+1}) - S_1(T_j)$, and $S_2(T_{j+1}) - S_2(T_j)$ are mutually independent. This is because regardless of what the agent observes in the first $j$ batches, i.e. $\mathcal{H}_{T_j}$, she is going to record rewards from both arms only $2^{j-1}$ numbers times in the $j + 1^{th}$ batch and it does not matter at which time indices these are recorded since all the future rewards from any arm are i.i.d. as well. This leads to

$$
\begin{aligned}
&\mathbb{E}[\exp(\lambda(S_2(T_{j+1}) - S_1(T_{j+1}) + 2^j\Delta_2))] \\
&= \mathbb{E}[\exp(\lambda(S_2(T_j) - S_1(T_j) + 2^{j-1}\Delta_2)) \\
&\qquad\qquad \mathbb{E}[\exp(\lambda(S_2(T_{j+1}) - S_2(T_j) - S_1(T_j) + S_1(T_{j+1}) + 2^{j-1}\Delta_2)) | \mathcal{H}_{T_j}]] \quad (26)
\end{aligned}
$$

However from the earlier analysis and the fact that the $j + 1^{th}$ batch contains $2^{j-1}$ recorded rewards from each arm, we have

$$
\begin{aligned}
&\mathbb{E}[\exp(\lambda(S_2(T_{j+1}) - S_2(T_j) - S_1(T_j) + S_1(T_{j+1}) + 2^{j-1}\Delta_2)) | \mathcal{H}_{T_j}] \\
&= \mathbb{E}[\exp(\lambda(S_2(T_{j+1}) - S_2(T_j) - 2^{j-1}\mu_2))] \mathbb{E}[\exp(-\lambda(S_1(T_{j+1}) - S_1(T_j) - 2^{j-1}\mu_1))] \\
&\leq \exp(2^{j-3}\lambda^2) \quad (27)
\end{aligned}
$$

where the last inequality follows from (19) and the fact that the first $l$ rewards and $l + 1^{th}$ reward from the same arm are independent since the rewards are i.i.d. Note that each arm will be picked infinitely often since probability selecting any arm in any batch will be almost surely positive due to using Gaussian distribution to select arms. Finally, (26) and (27), along with the induction step, shows that

$$
\mathbb{E}[\exp(\lambda(S_2(T_{j-1}) - S_1(T_{j-1}) + 2^{j-2}\Delta_2))] \leq \exp(2^{j-4}\lambda^2) \quad (28)
$$

for any $j \geq 2$. This result leads to the following bound for any $\lambda \geq 0$ and $x \geq 2$:

$$
\begin{aligned}
\mathbb{P}\left(\frac{S_2(T_{j-1}) - S_1(T_{j-1})}{\sqrt{2+2^{j-1}}} \geq Q^{-1}\left(\frac{1}{x}\right)\right) &\leq \mathbb{P}\left(\frac{S_2(T_{j-1}) - S_1(T_{j-1}) + 2^{j-2}\Delta_2}{\sqrt{2+2^{j-1}}} \geq Q^{-1}\left(\frac{1}{x}\right)\right) \\
&\leq \exp(\lambda^2/8 - \lambda Q^{-1}(1/x))
\end{aligned}
$$

where the last inequality follows from the Chernoff bound. Since $Q^{-1}(1/x) \geq 0$ when $x \geq 2$, setting $\lambda = 4Q^{-1}(1/x)$ shows that

$$\mathbb{P}\Big(\frac{S_2(T_{j-1}) - S_1(T_{j-1})}{\sqrt{2 + 2^{j-1}}} \geq Q^{-1}\Big(\frac{1}{x}\Big)\Big) \leq \exp(-2(Q^{-1}(1/x))^2)$$

for any $x \geq 2$. Finally, by (21)

$$\mathbb{P}\Big(\frac{S_2(T_{j-1}) - S_1(T_{j-1})}{\sqrt{2 + 2^{j-1}}} \geq Q^{-1}\Big(\frac{1}{x}\Big)\Big) \leq \frac{1}{x^{8/3}} \tag{29}$$

if $x \geq x_0$. Putting (29) back into (25) leads to:

$$\mathbb{E}\Big[\frac{1}{\mathbb{P}(A_{T_{j-1}+1} = 1 | H_{T_{j-1}})}\Big] \leq x_0 + \int_{x_0}^{\infty} \frac{1}{x^{8/3}} dx \leq x_0 + 1,$$

which proves (11) since $x_0$ is independent of any system parameter.

We now prove (12). Similar to the earlier analysis, we can describe the probability of selecting the second arm as the sample from the second arm being bigger than the first arm's:

$$\mathbb{P}(A_{T_{j-1}+1} = 2) = \mathbb{P}\Big(\mathcal{N}\Big(\frac{S_2(T_{j-1}) - S_1(T_{j-1})}{1 + 2^{j-2}}, \frac{2}{1 + 2^{j-2}}\Big) \geq 0\Big)$$

$$= \mathbb{P}\Big(\mathcal{N}\Big(\frac{S_2(T_{j-1}) - S_1(T_{j-1})}{1 + 2^{j-2}} + \frac{2^{j-2}}{1 + 2^{j-2}}\Delta_2, \frac{2}{1 + 2^{j-2}}\Big) \geq \frac{2^{j-2}}{1 + 2^{j-2}}\Delta_2\Big)$$

In view of (28), we know that $S_2(T_{j-1}) - S_1(T_{j-1}) + 2^{j-2}\Delta_2$ is sub Gaussian with variance proxy $2^{j-3}$. As a result, $\mathcal{N}(\frac{S_2(T_{j-1})-S_1(T_{j-1})}{1+2^{j-2}} + \frac{2^{j-2}}{1+2^{j-2}}\Delta_2, \frac{2}{1+2^{j-2}})$ has the following variance proxy:

$$\frac{2^{j-3}}{(1 + 2^{j-2})^2} + \frac{2}{1 + 2^{j-2}} = \frac{2^{j-3} + 2(1 + 2^{j-2})}{(1 + 2^{j-2})^2}.$$

This observation and Chernoff bound, which states that $\mathbb{P}(X \geq x) \leq \exp(-x^2/(2\sigma^2))$ if $x \geq 0$ and $X$ is sub Gaussian with variance proxy $\sigma^2$, lead to:

$$\mathbb{P}(A_{T_{j-1}+1} = 2) \leq \exp\Big(-\frac{(1 + 2^{j-2})^2}{2^{j-2} + 4(1 + 2^{j-2})} \frac{(2^{j-2})^2}{(1 + 2^{j-2})^2}\Delta_2^2\Big)$$

$$= \exp\Big(-\frac{2^{2j-4}}{2^{j-2} + 4(1 + 2^{j-2})}\Delta_2^2\Big)$$

$$\leq \exp\Big(-\frac{2^{2j-4}}{3 \times 2^j}\Delta_2^2\Big)$$

$$= \exp\Big(-\frac{2^{j-4}}{3}\Delta_2^2\Big)$$

which finishes the proof of (12).

# D    Results Related to Theorem 1

## D.1    Martingale Lemma

In this part, we present a key martingale lemma.

**Lemma 4.** *Let* $\mathcal{F}_t = \{Y_{A_1,1}, Y_{A_1,1}, ..., Y_{A_{T_b(t)},T_{b(t)}}, A_1, A_2, ..., A_t\}$*, then* $X_t = \exp(\lambda(S_i(T_{b(t)}) - \mu_i M_i(T_{b(t)})) - \frac{\lambda^2}{8}(1 + M_i(T_{b(t)})))$ *is a non-negative supermartingale adapted to* $\mathcal{F}_t$ *for any real* $\lambda$ *and* $i \in \{1, 2, .., K\}$*. Finally for any t we have*

$$\mathbb{E}[X_t] \leq 1, \tag{30}$$

*and in particular any stopping time* $\tau \leq \infty$ *for* $\{\mathcal{F}_t\}$ *satisfies the following inequality*

$$\mathbb{E}[X_\tau] \leq 1, \tag{31}$$

*where* $\lim_{t \to \infty} X_t = X_\infty$.

*Proof.* First of all, it is clear that $X_t$'s are integrable, since the rewards $\{Y_{A_t,t}\}$ are bounded. That means the only thing we need to prove is that $\{X_t\}$ is a supermartingale sequence, i.e. the following inequality

$$\mathbb{E}[X_{t+1}|\mathcal{F}_t] \leq X_t \tag{32}$$

almost surely for any positive integer $t$. By the definition $b(t)$ we know that $b(t)$ and $b(t+1)$ are functions of $\{A_1, A_2, ..., A_t\} \subseteq \mathcal{F}_t$. Note that the batch end points are decided by the actions taken. In addition, on $\{b(t) = b(t+1)\}$ $X_t$ is equal to $X_{t+1}$. These observations lead to

$$\mathbb{E}[X_{t+1}|\mathcal{F}_t] = \mathbb{E}[\exp(\lambda(S_i(T_{b(t+1)}) - \mu_i M_i(T_{b(t+1)})) - \frac{\lambda^2}{8}(1 + M_i(T_{b(t+1)})))|\mathcal{F}_t]$$

$$= \mathbb{1}(b(t) = b(t+1))X_t$$

$$+ \mathbb{E}[\mathbb{1}(b(t) \neq b(t+1))\exp(\sum_{j=T_{b(t)}+1}^{T_{b(t+1)}} \mathbb{1}(A_j = i, j = C_{b,k} \text{ or } C_{e,k} \text{ for some } k)(\lambda Y_{i,j} - \lambda\mu_i - \frac{\lambda^2}{8}))|\mathcal{F}_t]X_t \tag{33}$$

where the last equality follows from the definitions of $S_i(T_{b(t)})$ and $M_i(T_{b(t)})$. Note that $b(t) \neq b(t+1)$ if and only if $t$ is a batch end point, which leads to

$$\mathbb{1}(b(t) \neq b(t+1)) = \sum_{l=0}^{t-1} \mathbb{1}(T_{b(t+1)} = t, T_{b(t)} = l). \tag{34}$$

We first prove that $\mathbb{1}(T_{b(t+1)} = t, T_{b(t)} = l)$ and $\{Y_{i,j}\}_{j=l+1}^{t}$ are independent. To that end, it is enough to show that

$$\mathbb{P}(T_{b(t+1)} = t, T_{b(t)} = l, (Y_{i,l+1}, ..., Y_{i,t}) \in \mathcal{S}) = \mathbb{P}(T_{b(t+1)} = t, T_{b(t)} = l)\mathbb{P}((Y_{i,l+1}, ..., Y_{i,t}) \in \mathcal{S})$$

for any Borel set $\mathcal{S}$ of $\mathbb{R}^{t-l}$. Note that $T_{b(t)} = l$ and $T_{b(t+1)} = t$ if and only if $l$ is a batch end point, call this event $E_1$, and the smallest batch end point strictly bigger than $l$ is $t$, call this event $E_2$. This means $\{T_{b(t)} = l, T_{b(t+1)} = t\} = E_1 \cap E_2$. Then we know $\mathbb{P}((Y_{i,l+1}, ..., Y_{i,t}) \in \mathcal{S}, E_1) = \mathbb{P}((Y_{i,l+1}, ..., Y_{i,t}) \in \mathcal{S})\mathbb{P}(E_1)$ due to the fact that whether or not $l$ is a batch end point depends on the past actions $\{A_j\}_{j=1}^{l}$, which are independent of the future rewards from the $i^{th}$ arm $\{Y_{i,j}\}_{j=l+1}^{t}$. In addition, conditioned on the fact that there is a batch end point at $l$, i.e. the event $E_1$, different values for $\{Y_{i,j}\}_{j=l+1}^{t}$ won't change the probability of $E_2$ happening. This is because the agent can not use the information present in $\{Y_{i,j}\}_{j=l+1}^{t}$ unless the current that started at $l+1$ ends. As a result, we have $\mathbb{P}(E_2|E_1, (Y_{i,l+1}, ..., Y_{i,t}) \in \mathcal{S}) = \mathbb{P}(E_2|E_1)$. The overall analysis leads to

$$\mathbb{P}(T_{b(t+1)} = t, T_{b(t)} = l, (Y_{i,l+1}, ..., Y_{i,t}) \in \mathcal{S}) = \mathbb{P}(E_1, E_2, (Y_{i,l+1}, ..., Y_{i,t}) \in \mathcal{S})$$

$$= \mathbb{P}((Y_{i,l+1}, ..., Y_{i,t}) \in \mathcal{S})\mathbb{P}(E_1|(Y_{i,l+1}, ..., Y_{i,t}) \in \mathcal{S})\mathbb{P}(E_2|E_1, (Y_{i,l+1}, ..., Y_{i,t}) \in \mathcal{S})$$

$$= \mathbb{P}((Y_{i,l+1}, ..., Y_{i,t}) \in \mathcal{S})\mathbb{P}(E_1)\mathbb{P}(E_2|E_1)$$

$$= \mathbb{P}((Y_{i,l+1}, ..., Y_{i,t}) \in \mathcal{S})\mathbb{P}(E_1, E_2)$$

$$= \mathbb{P}((Y_{i,l+1}, ..., Y_{i,t}) \in \mathcal{S})\mathbb{P}(T_{b(t+1)} = t, T_{b(t)} = l), \tag{35}$$

which finishes the proof of the fact that $\mathbb{1}(T_{b(t+1)} = t, T_{b(t)} = l)$ and $\{Y_{i,j}\}_{j=l+1}^{t}$ are independent. Similar to the earlier analysis, conditioned on $\{T_{b(t+1)} = t, T_{b(t)} = l\}$, $\{A_1, Y_{A_1,1}, A_2, Y_{A_2,2}, ..., A_{T_{b(t)}}, Y_{A_{T_{b(t)}}, T_{b(t)}}\}$ and $\{Y_{i,j}\}_{j=l+1}^{t}$ are independent, because the future rewards from the $i^{th}$ arm can not affect the past observations, i.e. actions and rewards. In addition, conditioned on $\{T_{b(t+1)} = t, T_{b(t)} = l\}$, we know that the actions $\{A_j\}_{j=T_{b(t)}+1}^{t}$ are sampled according to the information present in $\{A_1, Y_{A_1,1}, A_2, Y_{A_2,2}, ..., A_{T_{b(t)}}, Y_{A_{T_{b(t)}}, T_{b(t)}}\}$. As a result, conditioned on $\{T_{b(t+1)} = t, T_{b(t)} = l\}$, $\mathcal{F}_t$ and $\{Y_{i,j}\}_{j=l+1}^{t}$ are independent. This overall analysis shows that for any Borel set $\mathcal{S}$ of $\mathbb{R}^{t-l}$ and any element $\mathcal{G}$ of the sigma algebra generated by $\mathcal{F}_t$ we have

$$\mathbb{P}(T_{b(t+1)} = t, T_{b(t)} = l, (Y_{i,l+1}, ..., Y_{i,t}) \in \mathcal{S}, \mathcal{G})$$

$$= \mathbb{P}(T_{b(t+1)} = t, T_{b(t)} = l, (Y_{i,l+1}, ..., Y_{i,t}) \in \mathcal{S})\mathbb{P}(\mathcal{G}|T_{b(t+1)} = t, T_{b(t)} = l, (Y_{i,l+1}, ..., Y_{i,t}) \in \mathcal{S})$$

$$= \mathbb{P}((Y_{i,l+1}, ..., Y_{i,t}) \in \mathcal{S})\mathbb{P}(T_{b(t+1)} = t, T_{b(t)} = l)\mathbb{P}(\mathcal{G}|T_{b(t+1)} = t, T_{b(t)} = l) \tag{36}$$

$$= \mathbb{P}((Y_{i,l+1}, ..., Y_{i,t}) \in \mathcal{S})\mathbb{P}(T_{b(t+1)} = t, T_{b(t)} = l, \mathcal{G}), \tag{37}$$

where (36) follows from (35) and the fact that conditioned on $\{T_{b(t+1)} = t, T_{b(t)} = l\}$, $\mathcal{F}_t$ and $\{Y_{i,j}\}_{j=l+1}^{t}$ are independent. Since $\mathcal{S}$ is arbitrary, $\{Y_{i,j}\}_{j=l+1}^{t}$ and $\mathbb{1}(T_{b(t+1)} = t, T_{b(t)} = l, \mathcal{G})$ are independent.

Now we go back to (34), and note that $\mathbb{1}(T_{b(t+1)} = t, T_{b(t)} = l)$ can be written as a sum of the terms of the following form

$$\mathbb{1}(T_{b(t+1)} = t, T_{b(t)} = l) \prod_{n=l+1}^{t} (\mathbb{1}(a_n = -1) + a_n \times \mathbb{1}(A_n = i, n = C_{b,k} \text{ or } C_{e,k} \text{ for some } k)), \tag{38}$$

where $a_n \in \{-1, 1\}$. Note that the terms of this form are indicator functions with disjoint domains. Then we have for any element $\hat{\mathcal{G}}$ of the sigma algebra generated by $\mathcal{F}_t$

$$\mathbb{E}[\mathbb{1}(T_{b(t+1)} = t, T_{b(t)} = l)( \prod_{n=l+1}^{t} (\mathbb{1}(a_n = -1) + a_n \times \mathbb{1}(A_n = i, n = C_{b,k} \text{ or } C_{e,k} \text{ for some } k))) \, \mathbb{1}(\hat{\mathcal{G}})$$

$$\exp( \sum_{j=T_{b(t)}+1}^{T_{b(t+1)}} \mathbb{1}(A_j = i, j = C_{b,k} \text{ or } C_{e,k} \text{ for some } k)(\lambda Y_{i,j} - \lambda \mu_i - \frac{\lambda^2}{8}))]$$

$$= \mathbb{E}[\mathbb{1}(T_{b(t+1)} = t, T_{b(t)} = l)( \prod_{n=l+1}^{t} (\mathbb{1}(a_n = -1) + a_n \times \mathbb{1}(A_n = i, n = C_{b,k} \text{ or } C_{e,k} \text{ for some } k))) \, \mathbb{1}(\hat{\mathcal{G}})$$

$$\exp( \sum_{j=l+1}^{t} \mathbb{1}(a_j = 1)(\lambda Y_{i,j} - \lambda \mu_i - \frac{\lambda^2}{8}))] \tag{39}$$

Note that $a_n$'s are deterministic variables and by the earlier analysis, i.e. (37), we know that $\mathbb{1}(T_{b(t+1)} = t, T_{b(t)} = l)(\prod_{n=l+1}^{t}(\mathbb{1}(a_n = -1) + a_n \times \mathbb{1}(A_n = i, n = C_{b,k} \text{ or } C_{e,k} \text{ for some } k))) \, \mathbb{1}(\hat{\mathcal{G}})$ and $\{Y_{i,j}\}_{j=l+1}^{t}$ are independent. Then by the fact that $\mathbb{E}[\exp(\lambda Y_{i,j} - \lambda \mu_i - \frac{\lambda^2}{8})] \leq 1$ due to (19), we have

$$\mathbb{E}[\mathbb{1}(T_{b(t+1)} = t, T_{b(t)} = l)( \prod_{n=l+1}^{t} (\mathbb{1}(a_n = -1) + a_n \times \mathbb{1}(A_n = i, n = C_{b,k} \text{ or } C_{e,k} \text{ for some } k))) \, \mathbb{1}(\hat{\mathcal{G}})$$

$$\exp( \sum_{j=T_{b(t)}+1}^{T_{b(t+1)}} \mathbb{1}(A_j = i, j = C_{b,k} \text{ or } C_{e,k} \text{ for some } k)(\lambda Y_{i,j} - \lambda \mu_i - \frac{\lambda^2}{8}))]$$

$$\leq \mathbb{E}[\mathbb{1}(T_{b(t+1)} = t, T_{b(t)} = l)( \prod_{n=l+1}^{t} (\mathbb{1}(a_n = -1) + a_n \times \mathbb{1}(A_n = i, n = C_{b,k} \text{ or } C_{e,k} \text{ for some } k))) \, \mathbb{1}(\hat{\mathcal{G}})].$$

Since $\hat{\mathcal{G}}$ is arbitrary, the previous inequality shows that

$$\mathbb{E}[\mathbb{1}(T_{b(t+1)} = t, T_{b(t)} = l)( \prod_{n=l+1}^{t} (\mathbb{1}(a_n = -1) + a_n \times \mathbb{1}(A_n = i, n = C_{b,k} \text{ or } C_{e,k} \text{ for some } k)))$$

$$\exp( \sum_{j=T_{b(t)}+1}^{T_{b(t+1)}} \mathbb{1}(A_j = i, j = C_{b,k} \text{ or } C_{e,k} \text{ for some } k)(\lambda Y_{i,j} - \lambda \mu_i - \frac{\lambda^2}{8})) | \mathcal{F}_t]$$

$$\leq \mathbb{1}(T_{b(t+1)} = t, T_{b(t)} = l)( \prod_{n=l+1}^{t} (\mathbb{1}(a_n = -1) + a_n \times \mathbb{1}(A_n = i, n = C_{b,k} \text{ or } C_{e,k} \text{ for some } k)))$$

almost surely, which leads to

$$\mathbb{E}[\mathbb{1}(T_{b(t+1)} = t, T_{b(t)} = l) \exp( \sum_{j=T_{b(t)}+1}^{T_{b(t+1)}} \mathbb{1}(A_j = i, j = C_{b,k} \text{ or } C_{e,k} \text{ for some } k)(\lambda Y_{i,j} - \lambda \mu_i - \frac{\lambda^2}{8})) | \mathcal{F}_t]$$

$$\leq \mathbb{1}(T_{b(t+1)} = t, T_{b(t)} = l) \tag{40}$$

almost surely by the observation in (38). Finally combining (34) and (40) proves that

$$\mathbb{E}[\mathbb{1}(b(t) \neq b(t+1)) \exp(\sum_{j=T_{b(t)}+1}^{T_{b(t+1)}} \mathbb{1}(A_j = i, j = C_{b,k} \text{ or } C_{e,k} \text{ for some } k)(\lambda Y_{i,j} - \lambda \mu_i - \frac{\lambda^2}{8}))| \mathcal{F}_t]$$

$$\leq \mathbb{1}(b(t) \neq b(t+1)) \tag{41}$$

almost surely. This inequality and (33) lead to (32). We have showed that $X_t$ is a supermartingale sequence.

Finally, we prove (30) and (31). Firstly, note that $T_{b(1)} = T_0 = 0$, $S_i(0) = 0$, and $M_i(0) = 0$, which lead to $\mathbb{E}[X_t] \leq \mathbb{E}[X_1] = \exp(-\lambda^2/8) \leq 1$ for any $t$ by the properties of supermartingales. Coupling this fact with the following theorem finishes the proof:

**Theorem 4.8.4 of [8].** *If $X_t$ is a non-negative supermartingale and $\tau \leq \infty$ is a stopping time, then $\mathbb{E}[X_\tau] \leq \mathbb{E}[X_1]$ where $\lim_{t \to \infty} X_t$ exists and $X_\infty = \lim_{t \to \infty} X_t$.*

$\square$

## D.2 Estimation Error Bound

In this section, we provide a proposition stating that if a certain arms is selected in sufficiently many cycles, then sample $\theta_i$ corresponding to that arm has to be close to the true mean with high probability.

**Proposition 5.** *Let $T \geq 2$, then for any positive integer $t$ and $i \in \{1, 2, ..., K\}$, we have*

$$\mathbb{P}\left(\theta_1(t) \leq \frac{\mu_1 + \mu_i}{2}, M_1(T_{b(t)}) \geq 32\sigma^2 \frac{\log(T)}{\Delta_i^2}\right) \leq \frac{2}{T} \tag{42}$$

*and*

$$\mathbb{P}\left(\theta_i(t) > \frac{\mu_1 + \mu_i}{2}, M_i(T_{b(t)}) \geq 32\sigma^2 \frac{\log(T)}{\Delta_i^2}\right) \leq \frac{2}{T}. \tag{43}$$

*Proof.* We first prove (42). Here we have

$$\mathbb{P}\left(\theta_1(t) \leq \frac{\mu_1 + \mu_i}{2}, M_1(T_{b(t)}) \geq 32\sigma^2 \frac{\log(T)}{\Delta_i^2}\right)$$

$$\leq \mathbb{P}\left(\theta_1(t) \leq \frac{\mu_1 + \mu_i}{2}, \frac{S_1(T_{b(t)})}{1 + M_1(T_{b(t)})} \geq \frac{3\mu_1 + \mu_i}{4}, M_1(T_{b(t)}) \geq 32\sigma^2 \frac{\log(T)}{\Delta_i^2}\right)$$

$$+ \mathbb{P}\left(\frac{S_1(T_{b(t)})}{1 + M_1(T_{b(t)})} < \frac{3\mu_1 + \mu_i}{4}, M_1(T_{b(t)}) \geq 32\sigma^2 \frac{\log(T)}{\Delta_i^2}\right) \tag{44}$$

We know that conditioned on $\mathcal{H}_{t-1}$, $\theta_1(t)$ is distributed as $\mathcal{N}(\frac{S_1(T_{b(t)})}{1+M_1(T_{b(t)})}, \frac{\sigma^2}{1+M_1(T_{b(t)})})$. This fact leads to

$$\mathbb{P}(\theta_1(t) \leq \frac{\mu_1 + \mu_i}{2} | \mathcal{H}_{t-1}) = Q\left(\frac{\sqrt{1 + M_1(T_{b(t)})}}{\sigma}\left(\frac{S_1(T_{b(t)})}{1 + M_1(T_{b(t)})} - \frac{\mu_1 + \mu_i}{2}\right)\right).$$

Then we have

$$\mathbb{P}\left(\theta_1(t) \leq \frac{\mu_1 + \mu_i}{2}, \frac{S_1(T_{b(t)})}{1 + M_1(T_{b(t)})} \geq \frac{3\mu_1 + \mu_i}{4}, M_1(T_{b(t)}) \geq 32\sigma^2 \frac{\log(T)}{\Delta_i^2}\right)$$

$$= \mathbb{E}[\mathbb{P}(\theta_1(t) \leq \frac{\mu_1 + \mu_i}{2} | \mathcal{H}_{t-1}) \mathbb{1}\left(\frac{S_1(T_{b(t)})}{1 + M_1(T_{b(t)})} \geq \frac{3\mu_1 + \mu_i}{4}, M_1(T_{b(t)}) \geq 32\sigma^2 \frac{\log(T)}{\Delta_i^2}\right)]$$

$$= \mathbb{E}[Q\left(\frac{\sqrt{1 + M_1(T_{b(t)})}}{\sigma}\left(\frac{S_1(T_{b(t)})}{1 + M_1(T_{b(t)})} - \frac{\mu_1 + \mu_i}{2}\right)\right)$$

$$\mathbb{1}\left(\frac{S_1(T_{b(t)})}{1 + M_1(T_{b(t)})} \geq \frac{3\mu_1 + \mu_i}{4}, M_1(T_{b(t)}) \geq 32\sigma^2 \frac{\log(T)}{\Delta_i^2}\right)] \tag{45}$$

where the first equality follows from the fact that $S_1(T_{b(t)})$ and $M_1(T_{b(t)})$ are measurable with respect to $\mathcal{H}_{t-1}$. Since

$$\frac{\sqrt{1 + M_1(T_{b(t)})}}{\sigma}\left(\frac{S_1(T_{b(t)})}{1 + M_1(T_{b(t)})} - \frac{\mu_1 + \mu_i}{2}\right) \geq \sqrt{2\log(T)}$$

on $\{\frac{S_1(T_{b(t)})}{1 + M_1(T_{b(t)})} \geq \frac{3\mu_1 + \mu_i}{4}, M_1(T_{b(t)}) \geq 32\sigma^2\frac{\log(T)}{\Delta_i^2}\}$, (45) leads to

$$\mathbb{P}\left(\theta_1(t) \leq \frac{\mu_1 + \mu_i}{2}, \frac{S_1(T_{b(t)})}{1 + M_1(T_{b(t)})} \geq \frac{3\mu_1 + \mu_i}{4}, M_1(T_{b(t)}) \geq 32\sigma^2\frac{\log(T)}{\Delta_i^2}\right) \leq Q(\sqrt{2\log(T)}).$$

However, we know that $Q(x) \leq \exp(-x^2/2)$ for $x \geq 1$ by (20), which results in

$$\mathbb{P}\left(\theta_1(t) \leq \frac{\mu_1 + \mu_i}{2}, \frac{S_1(T_{b(t)})}{1 + M_1(T_{b(t)})} \geq \frac{3\mu_1 + \mu_i}{4}, M_1(T_{b(t)}) \geq 32\sigma^2\frac{\log(T)}{\Delta_i^2}\right) \leq \frac{1}{T}. \qquad (46)$$

We now bound the second term on the right-hand side of (44). Note that

$$\mathbb{P}\left(\frac{S_1(T_{b(t)})}{1 + M_1(T_{b(t)})} < \frac{3\mu_1 + \mu_i}{4}, M_1(T_{b(t)}) \geq 32\sigma^2\frac{\log(T)}{\Delta_i^2}\right)$$

$$= \mathbb{P}\left(\frac{S_1(T_{b(t)}) - \mu_1 M_1(T_{b(t)})}{1 + M_1(T_{b(t)})} < \frac{3\mu_1 + \mu_i}{4} - \mu_1\frac{M_1(T_{b(t)})}{1 + M_1(T_{b(t)})}, M_1(T_{b(t)}) \geq 32\sigma^2\frac{\log(T)}{\Delta_i^2}\right)$$

$$= \mathbb{P}\left(\frac{S_1(T_{b(t)}) - \mu_1 M_1(T_{b(t)})}{1 + M_1(T_{b(t)})} < -\frac{\Delta_i}{4} + \mu_1\frac{1}{1 + M_1(T_{b(t)})}, M_1(T_{b(t)}) \geq 32\sigma^2\frac{\log(T)}{\Delta_i^2}\right). \qquad (47)$$

We know that $\mu_1 \leq 1$ and

$$\frac{\mu_1}{1 + M_1(T_{b(t)})} \leq \frac{\Delta_i^2}{32\sigma^2\log(T)} \leq \frac{\Delta_i}{8}$$

if $M_1(T_{b(t)}) \geq 32\sigma^2\frac{\log(T)}{\Delta_i^2}$. Note that the last inequality follows from the fact that $0 \leq \Delta_i \leq 1$ and $32\sigma^2\log(T) \geq 8$ for $T \geq 2$ and $\sigma^2 \geq 1$. This analysis and (47) indicate that

$$\mathbb{P}\left(\frac{S_1(T_{b(t)})}{1 + M_1(T_{b(t)})} < \frac{3\mu_1 + \mu_i}{4}, M_1(T_{b(t)}) \geq 32\sigma^2\frac{\log(T)}{\Delta_i^2}\right)$$

$$\leq \mathbb{P}\left(\frac{S_1(T_{b(t)}) - \mu_1 M_1(T_{b(t)})}{1 + M_1(T_{b(t)})} < -\frac{\Delta_i}{8}, M_1(T_{b(t)}) \geq 32\sigma^2\frac{\log(T)}{\Delta_i^2}\right). \qquad (48)$$

Then by Lemma 4 with $\lambda = -\frac{\Delta_i}{2}$

$$1 \geq \mathbb{E}[\exp(-\frac{\Delta_i}{2}(S_1(T_{b(t)}) - \mu_1 M_1(T_{b(t)})) - \frac{\Delta_i^2}{32}(1 + M_1(T_{b(t)})))$$

$$\mathbb{1}(\frac{S_1(T_{b(t)}) - \mu_1 M_1(T_{b(t)})}{1 + M_1(T_{b(t)})} < -\frac{\Delta_i}{8}, M_1(T_{b(t)}) \geq 32\sigma^2\frac{\log(T)}{\Delta_i^2})]$$

$$\geq \mathbb{E}[\exp(\frac{\Delta_i^2}{32}(1 + M_1(T_{b(t)}))) \mathbb{1}(\frac{S_1(T_{b(t)}) - \mu_1 M_1(T_{b(t)})}{1 + M_1(T_{b(t)})} < -\frac{\Delta_i}{8}, M_1(T_{b(t)}) \geq 32\sigma^2\frac{\log(T)}{\Delta_i^2})]$$

$$\qquad (49)$$

$$\geq T\mathbb{P}\left(\frac{S_1(T_{b(t)}) - \mu_1 M_1(T_{b(t)})}{1 + M_1(T_{b(t)})} < -\frac{\Delta_i}{8}, M_1(T_{b(t)}) \geq 32\sigma^2\frac{\log(T)}{\Delta_i^2}\right) \qquad (50)$$

where (49) from the condition set inside the indicator function, i.e. $\frac{S_1(T_{b(t)}) - \mu_1 M_1(T_{b(t)})}{1 + M_1(T_{b(t)})} < -\frac{\Delta_i}{8}$.
Similarly, the condition $M_1(T_{b(t)}) \geq 32\sigma^2\frac{\log(T)}{\Delta_i^2}$ and $\sigma^2 \geq 1$ lead to (50). Then combining (48) and (50) results in

$$\mathbb{P}\left(\frac{S_1(T_{b(t)})}{1 + M_1(T_{b(t)})} < \frac{3\mu_1 + \mu_i}{4}, M_1(T_{b(t)}) \geq 32\sigma^2\frac{\log(T)}{\Delta_i^2}\right) \leq \frac{1}{T}$$

which, along with (44) and (46), proves (42). We now prove (43). However its proof is almost the same as the proof of (42). Similarly we have

$$\mathbb{P}\left(\theta_i(t) > \frac{\mu_1 + \mu_i}{2}, M_i(T_{b(t)}) \geq 32\sigma^2 \frac{\log(T)}{\Delta_i^2}\right)$$

$$\leq \mathbb{P}\left(\theta_i(t) > \frac{\mu_1 + \mu_i}{2}, \frac{S_i(T_{b(t)})}{1 + M_i(T_{b(t)})} \leq \frac{\mu_1 + 3\mu_i}{4}, M_i(T_{b(t)}) \geq 32\sigma^2 \frac{\log(T)}{\Delta_i^2}\right)$$

$$+ \mathbb{P}\left(\frac{S_i(T_{b(t)})}{1 + M_i(T_{b(t)})} > \frac{\mu_1 + 3\mu_i}{4}, M_i(T_{b(t)}) \geq 32\sigma^2 \frac{\log(T)}{\Delta_i^2}\right) \tag{51}$$

Here conditioned on $\mathcal{H}_{t-1}$, $\theta_i(t)$ is distributed as $\mathcal{N}(\frac{S_i(T_{b(t)})}{1+M_i(T_{b(t)})}, \frac{\sigma^2}{1+M_i(T_{b(t)})})$. Then it is easy to see that

$$\mathbb{P}(\theta_i(t) > \frac{\mu_1 + \mu_i}{2} | \mathcal{H}_{t-1}) = Q\left(\frac{\sqrt{1 + M_i(T_{b(t)})}}{\sigma}\left(\frac{\mu_1 + \mu_i}{2} - \frac{S_i(T_{b(t)})}{1 + M_i(T_{b(t)})}\right)\right),$$

which will lead to

$$\mathbb{P}\left(\theta_i(t) > \frac{\mu_1 + \mu_i}{2}, \frac{S_i(T_{b(t)})}{1 + M_i(T_{b(t)})} \leq \frac{\mu_1 + 3\mu_i}{4}, M_i(T_{b(t)}) \geq 32\sigma^2 \frac{\log(T)}{\Delta_i^2}\right) \leq \frac{1}{T} \tag{52}$$

by an analysis that is almost the same as the one prior to (46). On the other hand, the last summand in (51) satisfies

$$\mathbb{P}\left(\frac{S_i(T_{b(t)})}{1 + M_i(T_{b(t)})} > \frac{\mu_1 + 3\mu_i}{4}, M_i(T_{b(t)}) \geq 32\sigma^2 \frac{\log(T)}{\Delta_i^2}\right)$$

$$= \mathbb{P}\left(\frac{S_i(T_{b(t)}) - \mu_i M_i(T_{b(t)})}{1 + M_i(T_{b(t)})} > \frac{\mu_1 + 3\mu_i}{4} - \mu_i \frac{M_i(T_{b(t)})}{1 + M_i(T_{b(t)})}, M_i(T_{b(t)}) \geq 32\sigma^2 \frac{\log(T)}{\Delta_i^2}\right)$$

$$\leq \mathbb{P}\left(\frac{S_i(T_{b(t)}) - \mu_i M_i(T_{b(t)})}{1 + M_i(T_{b(t)})} > \frac{\Delta_i}{8}, M_i(T_{b(t)}) \geq 32\sigma^2 \frac{\log(T)}{\Delta_i^2}\right).$$

Similar to the analysis in (50), setting $\lambda = \frac{\Delta_i}{2}$ in Lemma 4 leads to

$$\mathbb{P}\left(\frac{S_i(T_{b(t)})}{1 + M_i(T_{b(t)})} > \frac{\mu_1 + 3\mu_i}{4}, M_i(T_{b(t)}) \geq 32\sigma^2 \frac{\log(T)}{\Delta_i^2}\right) \leq \frac{1}{T}. \tag{53}$$

In view of (51), combining (52) and (53) finishes the proof of (43). □

### D.3 Bounds on Functions of $Q$

This section provides bounds on various functions of $Q$ function. Before we state our results, we introduce for any $i$

$$\hat{\tau}_{i,j} = \begin{cases} \min\{t \in \mathbb{Z}^+ | A_t = i, t = C_{b,k} \text{ or } C_{e,k} \text{ for some } k\} & \text{if } j = 1 \\ \min\{t \in \mathbb{Z}^+ | A_t = i, t = C_{b,k} \text{ or } C_{e,k} \text{ for some } k, t > \hat{\tau}_{i,j-1}\} & \text{if } j > 1. \end{cases}$$

Note that if a set is empty, then $\hat{\tau}_{i,j}$ is set to be infinity. Here $\hat{\tau}_{i,j}$ denotes the time index where we choose the $i^{th}$ arm for $j^{th}$ time at the beginning or at the end of a cycle. It is clear that $\hat{\tau}_{i,j}$ is a stopping time for $\{\mathcal{F}_t\}$ specified in Lemma 4, so (31) in Lemma 4 remains true if we set $\tau$ to be $\hat{\tau}_{i,j}$:

$$\mathbb{E}[\lambda(S_i(T_{b(\hat{\tau}_{i,j})}) - \mu_i M_i(T_{b(\hat{\tau}_{i,j})})) - \frac{\lambda^2}{8}(1 + M_i(T_{b(\hat{\tau}_{i,j})}))] \leq 1 \tag{54}$$

for any $i \in \{1, 2, ..., K\}$ and real $\lambda$.

**Lemma 6.** *For any $j > 1$ and $i \in \{1, 2, ..., K\}$, if $x \geq 0$:*

$$\mathbb{P}\left(\hat{\tau}_{i,j} < \infty, \frac{S_i(T_{b(\hat{\tau}_{i,j})}) - \mu_i M_i(T_{b(\hat{\tau}_{i,j})})}{\sqrt{1 + M_i(T_{b(\hat{\tau}_{i,j})})}} > x\right) \leq \exp(-2x^2/\alpha), \tag{55}$$

*and if $x \leq 0$:*

$$\mathbb{P}\left(\hat{\tau}_{i,j} < \infty, \frac{S_i(T_{b(\hat{\tau}_{i,j})}) - \mu_i M_i(T_{b(\hat{\tau}_{i,j})})}{\sqrt{1 + M_i(T_{b(\hat{\tau}_{i,j})})}} < x\right) \leq \exp(-2x^2/\alpha). \tag{56}$$

*Proof.* We start with the proof of (55). Assume $x \geq 0$. By setting $\lambda = \frac{4x}{\sqrt{\alpha j}}$ in (54), we have

$$1 \geq \mathbb{E}[\exp(\frac{4x}{\sqrt{\alpha j}}(S_i(T_{b(\hat{\tau}_{i,j})}) - \mu_i M_i(T_{b(\hat{\tau}_{i,j})})) - \frac{2x^2}{\alpha j}(1 + M_i(T_{b(\hat{\tau}_{i,j})})))]$$

$$\geq \mathbb{E}[\exp(\frac{4x}{\sqrt{\alpha j}}(S_i(T_{b(\hat{\tau}_{i,j})}) - \mu_i M_i(T_{b(\hat{\tau}_{i,j})})) - \frac{2x^2}{\alpha j}(1 + M_i(T_{b(\hat{\tau}_{i,j})})))$$

$$\mathbb{1}\left(\hat{\tau}_{i,j} < \infty, \frac{S_i(T_{b(\hat{\tau}_{i,j})}) - \mu_1 M_i(T_{b(\hat{\tau}_{i,j})})}{\sqrt{1 + M_i(T_{b(\hat{\tau}_{i,j})})}} > x\right)]$$

$$\geq \mathbb{E}[\exp(\frac{4x}{\sqrt{\alpha j}}(S_i(T_{b(\hat{\tau}_{i,j})}) - \mu_i M_i(T_{b(\hat{\tau}_{i,j})})) - \frac{2x^2}{\alpha}) \mathbb{1}(\hat{\tau}_{i,j} < \infty, \frac{S_i(T_{b(\hat{\tau}_{i,j})}) - \mu_1 M_i(T_{b(\hat{\tau}_{i,j})})}{\sqrt{1 + M_i(T_{b(\hat{\tau}_{i,j})})}} > x)]$$

$$(57)$$

Note that the last inequality follows from $M_i(T_{b(\hat{\tau}_{i,j})}) \leq j - 1$ since $T_{b(\hat{\tau}_{i,j})}$ is the last batch end point, which is strictly smaller than $\hat{\tau}_{i,j}$. Also by the construction of Algorithm 1, $\max\{1, \lceil \alpha M_i(T_{b(\hat{\tau}_{i,j})}) \rceil\} \geq j$. Given that $j > 1$, $M_i(T_{b(\hat{\tau}_{i,j})}) \geq 1$, which results in $\alpha(1 + M_i(T_{b(\hat{\tau}_{i,j})})) \geq j$ by $\alpha > 1$. Then on $\{\hat{\tau}_{i,j} < \infty, \frac{S_i(T_{b(\hat{\tau}_{i,j})}) - \mu_i M_i(T_{b(\hat{\tau}_{i,j})})}{\sqrt{1 + M_i(T_{b(\hat{\tau}_{i,j})})}} > x\}$, we have

$$\frac{4x}{\sqrt{\alpha j}}(S_i(T_{b(\hat{\tau}_{i,j})}) - \mu_i M_i(T_{b(\hat{\tau}_{i,j})})) \geq \frac{4x^2}{\alpha}.$$

Using this inequality inside (57) proves (55). Finally, the proof of (56) will follow the same steps with a single exception: here we set $\lambda = -\frac{4x}{\sqrt{\alpha j}}$ in (54). This finishes the proof.

$\square$

**Proposition 7.** *Assume $5\sigma^2/4 \geq \alpha$. Then for any positive integer $j$ and $i \geq 2$, we have*

$$\mathbb{E}\left[\mathbb{1}(\hat{\tau}_{1,j} < \infty)\frac{1}{Q^2\left(\frac{\sqrt{1+M_1(T_{b(\hat{\tau}_{1,j})})}}{\sigma}\left(\frac{\mu_1+\mu_i}{2} - \frac{S_1(T_{b(\hat{\tau}_{1,j})})}{1+M_1(T_{b(\hat{\tau}_{1,j})})}\right)\right)}\right] \leq C \qquad (58)$$

*and*

$$\mathbb{E}\left[\mathbb{1}(\hat{\tau}_{i,j} < \infty)\frac{1}{Q^2\left(\frac{\sqrt{1+M_i(T_{b(\hat{\tau}_{i,j})})}}{\sigma}\left(\frac{S_i(T_{b(\hat{\tau}_{i,j})})}{1+M_i(T_{b(\hat{\tau}_{i,j})})} - \frac{\mu_1+\mu_i}{2}\right)\right)}\right] \leq C \qquad (59)$$

*where $C$ is an absolute constant independent of the system parameters.*

*Proof.* We start with the proof of (58) and the proof of (59) will follow similarly. First note that if $j = 1$, then $M_1$ and $S_1$ terms will be zeros, and as a result, the denominator inside the expectation is lower bounded as follows:

$$Q(\frac{\mu_1 + \mu_i}{2\sigma}) \geq Q(0.5)$$

since $\mu_i, \mu_1 \leq 1$ and $\sigma^2 \geq 1$. So for $j = 1$, we can choose $C$ to be $1/Q^2(0.5)$.

For $j > 1$ by (22)

$$\mathbb{E}\left[\mathbb{1}(\hat{\tau}_{1,j} < \infty)\frac{1}{Q^2\left(\frac{\sqrt{1+M_1(T_{b(\hat{\tau}_{1,j})})}}{\sigma}\left(\frac{\mu_1+\mu_i}{2} - \frac{S_1(T_{b(\hat{\tau}_{1,j})})}{1+M_1(T_{b(\hat{\tau}_{1,j})})}\right)\right)}\right]$$

$$= \int_{x=0}^{\infty} \mathbb{P}\left(1(\hat{\tau}_{1,j} < \infty)\frac{1}{Q^2\left(\frac{\sqrt{1+M_1(T_{b(\hat{\tau}_{1,j})})}}{\sigma}\left(\frac{\mu_1+\mu_i}{2} - \frac{S_1(T_{b(\hat{\tau}_{1,j})})}{1+M_1(T_{b(\hat{\tau}_{1,j})})}\right)\right)} > x\right)dx$$

$$\leq 4 + \int_{x=4}^{\infty} \mathbb{P}\left(\hat{\tau}_{1,j} < \infty, Q\left(\frac{\sqrt{1+M_1(T_{b(\hat{\tau}_{1,j})})}}{\sigma}\left(\frac{\mu_1+\mu_i}{2} - \frac{S_1(T_{b(\hat{\tau}_{1,j})})}{1+M_1(T_{b(\hat{\tau}_{1,j})})}\right)\right) < \frac{1}{\sqrt{x}}\right)dx$$

$$\leq 4 + \int_{x=4}^{\infty} \mathbb{P}\left(\hat{\tau}_{1,j} < \infty, \frac{\sqrt{1+M_1(T_{b(\hat{\tau}_{1,j})})}}{\sigma}\left(\frac{\mu_1+\mu_i}{2} - \frac{S_1(T_{b(\hat{\tau}_{1,j})})}{1+M_1(T_{b(\hat{\tau}_{1,j})})}\right) > Q^{-1}(1/\sqrt{x})\right)dx,$$

$$\tag{60}$$

where the last inequality follows from the fact that $Q(\cdot)$ is a decreasing function. Now note that

$$\frac{\mu_1+\mu_i}{2} - \frac{S_1(T_{b(\hat{\tau}_{1,j})})}{1+M_1(T_{b(\hat{\tau}_{1,j})})} = \frac{\mu_1+\mu_i}{2} - \mu_1 + \mu_1 - \frac{\mu_1 M_1(T_{b(\hat{\tau}_{1,j})})}{1+M_1(T_{b(\hat{\tau}_{1,j})})} - \frac{S_1(T_{b(\hat{\tau}_{1,j})}) - \mu_1 M_1(T_{b(\hat{\tau}_{1,j})})}{1+M_1(T_{b(\hat{\tau}_{1,j})})}$$

$$\leq \frac{1}{1+M_1(T_{b(\hat{\tau}_{1,j})})} - \frac{S_1(T_{b(\hat{\tau}_{1,j})}) - \mu_1 M_1(T_{b(\hat{\tau}_{1,j})})}{1+M_1(T_{b(\hat{\tau}_{1,j})})}.$$

The last inequality follows from $\mu_1 \leq 1$. This analysis shows that if $\sqrt{1+M_1(T_{b(\hat{\tau}_{1,j})})}(\frac{\mu_1+\mu_i}{2} - \frac{S_1(T_{b(\hat{\tau}_{1,j})})}{1+M_1(T_{b(\hat{\tau}_{1,j})})}) > \sigma Q^{-1}(1/\sqrt{x})$, then $-\frac{S_1(T_{b(\hat{\tau}_{1,j})}) - \mu_1 M_1(T_{b(\hat{\tau}_{1,j})})}{\sqrt{1+M_1(T_{b(\hat{\tau}_{1,j})})}} > \sigma Q^{-1}(1/\sqrt{x}) - \frac{1}{\sqrt{1+M_1(T_{b(\hat{\tau}_{1,j})})}}$. Since $M_1(T_{b(\hat{\tau}_{1,j})}) \geq 0$ and $\sigma^2 \geq 1$, the last inequality leads to

$$\mathbb{E}\left[\mathbb{1}(\hat{\tau}_{1,j} < \infty)\frac{1}{Q^2\left(\frac{\sqrt{1+M_1(T_{b(\hat{\tau}_{1,j})})}}{\sigma}\left(\frac{\mu_1+\mu_i}{2} - \frac{S_1(T_{b(\hat{\tau}_{1,j})})}{1+M_1(T_{b(\hat{\tau}_{1,j})})}\right)\right)}\right]$$

$$\leq 4 + \int_{x=4}^{\infty} \mathbb{P}\left(\hat{\tau}_{1,j} < \infty, -\frac{S_1(T_{b(\hat{\tau}_{1,j})}) - \mu_1 M_1(T_{b(\hat{\tau}_{1,j})})}{\sqrt{1+M_1(T_{b(\hat{\tau}_{1,j})})}} > \sigma(Q^{-1}(1/\sqrt{x}) - 1)\right)dx \tag{61}$$

by (60). However, (21) indicates that there exists $x_0$ such that if $x \geq x_0$, then $Q^{-1}(\frac{1}{\sqrt{x}}) - 1 \geq \sqrt{\frac{16}{25}\log(x)}$, which leads to

$$\mathbb{P}\left(\hat{\tau}_{1,j} < \infty, -\frac{S_1(T_{b(\hat{\tau}_{1,j})}) - \mu_1 M_1(T_{b(\hat{\tau}_{1,j})})}{\sqrt{1+M_1(T_{b(\hat{\tau}_{1,j})})}} > \sigma(Q^{-1}(1/\sqrt{x}) - 1)\right) \leq \exp(-\frac{32\sigma^2}{25\alpha}\log(x))$$

by Lemma 6 for $x \geq x_0$. Since $\alpha \leq \frac{5\sigma^2}{4}$, the last inequality can be refined to

$$\mathbb{P}\left(\hat{\tau}_{1,j} < \infty, -\frac{S_1(T_{b(\hat{\tau}_{1,j})}) - \mu_1 M_1(T_{b(\hat{\tau}_{1,j})})}{\sqrt{1+M_1(T_{b(\hat{\tau}_{1,j})})}} > \sigma(Q^{-1}(1/\sqrt{x}) - 1)\right) \leq \frac{1}{x^{128/125}} \tag{62}$$

for any $x \geq x_0$. Finally, (61) and (62) shows that

$$\mathbb{E}\left[\mathbb{1}(\hat{\tau}_{1,j} < \infty)\frac{1}{Q^2\left(\frac{\sqrt{1+M_1(T_{b(\hat{\tau}_{1,j})})}}{\sigma}\left(\frac{\mu_1+\mu_i}{2} - \frac{S_1(T_{b(\hat{\tau}_{1,j})})}{1+M_1(T_{b(\hat{\tau}_{1,j})})}\right)\right)}\right] \leq x_0 + \int_{x=x_0}^{\infty} \frac{1}{x^{128/125}}dx$$

This result proves (58) since $x_0$ is an absolute constant independent of the system parameters and the final integral is finite. Note that we already upper bounded the $j = 1$ term earlier by an absolute constant so we can just take the maximum of the two.

The proof of (59) will similarly follow. First of all, for $j = 1$ we can bound the $Q$ terms inside the expectation as follows

$$Q(-\frac{\mu_1 + \mu_i}{2\sigma}) \geq Q(0) = 0.5$$

where these follow from the fact that expected means are non-negative. So for $j = 1$, we can choose $C$ to be 4. If $j > 1$, we have

$$\mathbb{E}\left[ \mathbb{1}(\hat{\tau}_{i,j} < \infty) \frac{1}{Q^2\left( \frac{\sqrt{1+M_i(T_{b(\hat{\tau}_{i,j})})}}{\sigma} \left( \frac{S_i(T_{b(\hat{\tau}_{i,j})})}{1+M_i(T_{b(\hat{\tau}_{i,j})})} - \frac{\mu_1+\mu_i}{2} \right) \right)} \right]$$

$$\leq 4 + \int_{x=4}^{\infty} \mathbb{P}\left( \hat{\tau}_{i,j} < \infty, \frac{\sqrt{1+M_i(T_{b(\hat{\tau}_{i,j})})}}{\sigma} \left( \frac{S_i(T_{b(\hat{\tau}_{i,j})})}{1+M_i(T_{b(\hat{\tau}_{i,j})})} - \frac{\mu_1+\mu_i}{2} \right) > Q^{-1}(1/\sqrt{x}) \right) dx.$$

$$(63)$$

As for the terms inside:

$$\frac{S_i(T_{b(\hat{\tau}_{i,j})})}{1+M_i(T_{b(\hat{\tau}_{i,j})})} - \frac{\mu_1+\mu_i}{2} = \frac{S_i(T_{b(\hat{\tau}_{i,j})}) - \mu_i M_i(T_{b(\hat{\tau}_{i,j})})}{1+M_i(T_{b(\hat{\tau}_{i,j})})} + \frac{\mu_i M_i(T_{b(\hat{\tau}_{i,j})})}{1+M_i(T_{b(\hat{\tau}_{i,j})})} - \mu_i + \mu_i - \frac{\mu_1+\mu_i}{2}$$

$$\leq \frac{S_i(T_{b(\hat{\tau}_{i,j})}) - \mu_i M_i(T_{b(\hat{\tau}_{i,j})})}{1+M_i(T_{b(\hat{\tau}_{i,j})})},$$

where the last step follows from $\mu_i \geq 0$. This analysis and (63) lead to

$$\mathbb{E}\left[ \mathbb{1}(\hat{\tau}_{i,j} < \infty) \frac{1}{Q^2\left( \frac{\sqrt{1+M_i(T_{b(\hat{\tau}_{i,j})})}}{\sigma} \left( \frac{S_i(T_{b(\hat{\tau}_{i,j})})}{1+M_i(T_{b(\hat{\tau}_{i,j})})} - \frac{\mu_1+\mu_i}{2} \right) \right)} \right]$$

$$\leq 4 + \int_{x=4}^{\infty} \mathbb{P}\left( \hat{\tau}_{i,j} < \infty, \frac{S_i(T_{b(\hat{\tau}_{i,j})}) - \mu_i M_i(T_{b(\hat{\tau}_{i,j})})}{\sqrt{1+M_i(T_{b(\hat{\tau}_{i,j})})}} > \sigma Q^{-1}(1/\sqrt{x}) \right) dx.$$

The rest of the proof follows exactly the same way it did after (61) in the proof of (58). $\qquad\square$

**Proposition 8.** *Assume $5\sigma^2/4 \geq \alpha$, $T \geq 2$, and $j > 1024\alpha\sigma^2 \frac{\log(T)}{\Delta_i^2}$, then*

$$\mathbb{E}\left[ \mathbb{1}(\hat{\tau}_{1,j} < \infty) \left( \frac{1}{Q\left( \frac{\sqrt{1+M_1(T_{b(\hat{\tau}_{1,j})})}}{\sigma} \left( \frac{\mu_1+\mu_i}{2} - \frac{S_1(T_{b(\hat{\tau}_{1,j})})}{1+M_1(T_{b(\hat{\tau}_{1,j})})} \right) \right)} - 1 \right)^2 \right] \leq \frac{C}{T}$$

*where $C$ is an absolute constant independent of the system parameters.*

*Proof.* Since $j > 1$, we know that $j \leq \lceil \alpha \times M_1(T_{b(\hat{\tau}_{1,j})}) \rceil$ by the construction of Algorithm 1, and as a result,

$$1024\alpha\sigma^2 \frac{\log(T)}{\Delta_i^2} < j \leq \lceil \alpha \times M_1(T_{b(\hat{\tau}_{1,j})}) \rceil \leq 2\alpha M_1(T_{b(\hat{\tau}_{1,j})}),$$

which leads to $M_1(T_{b(\hat{\tau}_{1,j})}) > 512\sigma^2 \frac{\log(T)}{\Delta_i^2}$. Note that $M_1(T_{b(\hat{\tau}_{1,j})}) \geq 1$ if $j > 1$. Then by (22) we have

$$\mathbb{E}\left[\mathbb{1}(\hat{\tau}_{1,j} < \infty)\left(\frac{1}{Q\left(\frac{\sqrt{1+M_1(T_{b(\hat{\tau}_{1,j})})}}{\sigma}\left(\frac{\mu_1+\mu_i}{2} - \frac{S_1(T_{b(\hat{\tau}_{1,j})})}{1+M_1(T_{b(\hat{\tau}_{1,j})})}\right)\right)} - 1\right)^2\right]$$

$$= \int_{x=0}^{\infty} \mathbb{P}\left(\mathbb{1}(\hat{\tau}_{1,j} < \infty)\left(\frac{1}{Q\left(\frac{\sqrt{1+M_1(T_{b(\hat{\tau}_{1,j})})}}{\sigma}\left(\frac{\mu_1+\mu_i}{2} - \frac{S_1(T_{b(\hat{\tau}_{1,j})})}{1+M_1(T_{b(\hat{\tau}_{1,j})})}\right)\right)} - 1\right)^2 > x\right)dx$$

$$= \int_{x=0}^{\infty} \mathbb{P}\left(\hat{\tau}_{1,j} < \infty, Q\left(\frac{\sqrt{1+M_1(T_{b(\hat{\tau}_{1,j})})}}{\sigma}\left(\frac{\mu_1+\mu_i}{2} - \frac{S_1(T_{b(\hat{\tau}_{1,j})})}{1+M_1(T_{b(\hat{\tau}_{1,j})})}\right)\right) < \frac{1}{1+\sqrt{x}}\right)dx$$

$$= \int_{x=0}^{\infty} \mathbb{P}\left(\hat{\tau}_{1,j} < \infty, \frac{\sqrt{1+M_1(T_{b(\hat{\tau}_{1,j})})}}{\sigma}\left(\frac{\mu_1+\mu_i}{2} - \frac{S_1(T_{b(\hat{\tau}_{1,j})})}{1+M_1(T_{b(\hat{\tau}_{1,j})})}\right) > Q^{-1}(\frac{1}{1+\sqrt{x}})\right)dx.$$
(64)

These steps follow from simple algebra and the decreasing nature of $Q(\cdot)$.

Now the goal here is to divide the integral in (64) into three regions, where the the contribution from each is in the order of $1/T$. As we will show next, the first region $[0, 1/T]$ will be upper bounded by $1/T$. In the second region $[1/T, x_0]$ for some big $x_0$, the probability term will be of $O(1/T)$. Finally, the integrand in the third region $[x_0, \infty)$ will decay faster than $\frac{1}{Tx^{\delta}}$ for some $\delta > 1$, and this will result in a contribution of order $O(1/T)$. The fact that $x_0$ is an absolute constant will finish the proof. We will start the analysis with the second region. First of all, the symmetry of the normal distribution leads to

$$Q^{-1}(\frac{1}{1+\sqrt{x}}) = -Q^{-1}(\frac{\sqrt{x}}{1+\sqrt{x}})$$
(65)

for $x \geq 0$. Given that $Q(y) \leq \exp(-y^2/2)$ for $y \geq 0$ by the Chernoff bound, we also know $Q^{-1}(\exp(-y^2/2)) \leq y$ for $y \geq 0$. Clearly $\sqrt{2\log(1+\sqrt{T})} \geq 0$ and letting $y = \sqrt{2\log(1+\sqrt{T})}$ shows that

$$Q^{-1}(\frac{\sqrt{x}}{1+\sqrt{x}}) \leq \sqrt{2\log(1+\sqrt{T})}$$

if $x = 1/T$. Since $Q^{-1}(\frac{1}{1+\sqrt{x}})$ is an increasing function of $x$, for $x \geq 1/T$, we have

$$Q^{-1}(\frac{1}{1+\sqrt{x}}) \geq -\sqrt{2\log(1+\sqrt{T})}.$$

by (65). This analysis proves that

$$Q^{-1}(\frac{1}{1+\sqrt{x}}) + 2\sqrt{2\log(T)} \geq 0$$
(66)

for $x \geq 1/T$ since $T \geq 2$. In addition, by simple algebra

$$\frac{\sqrt{1+M_1(T_{b(\hat{\tau}_{1,j})})}}{\sigma}\left(\frac{\mu_1+\mu_i}{2} - \frac{S_1(T_{b(\hat{\tau}_{1,j})})}{1+M_1(T_{b(\hat{\tau}_{1,j})})}\right)$$

$$= \frac{\sqrt{1+M_1(T_{b(\hat{\tau}_{1,j})})}}{\sigma}\left(\frac{\mu_1+\mu_i}{2} - \mu_1 + \mu_1 - \frac{\mu_1 M_1(T_{b(\hat{\tau}_{1,j})})}{1+M_1(T_{b(\hat{\tau}_{1,j})})} - \frac{S_1(T_{b(\hat{\tau}_{1,j})}) - \mu_1 M_1(T_{b(\hat{\tau}_{1,j})})}{1+M_1(T_{b(\hat{\tau}_{1,j})})}\right)$$

$$= \frac{\mu_1 M_1(T_{b(\hat{\tau}_{1,j})}) - S_1(T_{b(\hat{\tau}_{1,j})})}{\sigma\sqrt{1+M_1(T_{b(\hat{\tau}_{1,j})})}} + \frac{\sqrt{1+M_1(T_{b(\hat{\tau}_{1,j})})}}{\sigma}\left(-\frac{\Delta_i}{2} + \frac{\mu_1}{1+M_1(T_{b(\hat{\tau}_{1,j})})}\right). \quad (67)$$

Note here that $\frac{\mu_1}{1+M_1(T_{b(\hat{\tau}_{1,j})})} \leq \frac{\Delta_i^2}{512\sigma^2 \log(T)} \leq \frac{\Delta_i}{4}$ since $M_1(T_{b(\hat{\tau}_{1,j})}) > 512\sigma^2 \frac{\log(T)}{\Delta_i^2}$, $T \geq 2$, and $\sigma^2 \geq 1$. Then (67) leads to

$$\frac{\sqrt{1 + M_1(T_{b(\hat{\tau}_{1,j})})}}{\sigma}\left(\frac{\mu_1 + \mu_i}{2} - \frac{S_1(T_{b(\hat{\tau}_{1,j})})}{1 + M_1(T_{b(\hat{\tau}_{1,j})})}\right)$$

$$\leq \frac{\mu_1 M_1(T_{b(\hat{\tau}_{1,j})}) - S_1(T_{b(\hat{\tau}_{1,j})})}{\sigma\sqrt{1 + M_1(T_{b(\hat{\tau}_{1,j})})}} - \frac{\Delta_i\sqrt{1 + M_1(T_{b(\hat{\tau}_{1,j})})}}{4\sigma}$$

$$\leq \frac{\mu_1 M_1(T_{b(\hat{\tau}_{1,j})}) - S_1(T_{b(\hat{\tau}_{1,j})})}{\sigma\sqrt{1 + M_1(T_{b(\hat{\tau}_{1,j})})}} - 4\sqrt{2\log(T)}. \tag{68}$$

where the last inequality is the result of $M_1(T_{b(\hat{\tau}_{1,j})}) > 512\sigma^2 \frac{\log(T)}{\Delta_i^2}$. The overall analysis shows that if $x \geq 1/T$:

$$\mathbb{P}\left(\hat{\tau}_{1,j} < \infty, \frac{\sqrt{1 + M_1(T_{b(\hat{\tau}_{1,j})})}}{\sigma}\left(\frac{\mu_1 + \mu_i}{2} - \frac{S_1(T_{b(\hat{\tau}_{1,j})})}{1 + M_1(T_{b(\hat{\tau}_{1,j})})}\right) > Q^{-1}(\frac{1}{1+\sqrt{x}})\right)$$

$$\leq \mathbb{P}\left(\hat{\tau}_{1,j} < \infty, \frac{\mu_1 M_1(T_{b(\hat{\tau}_{1,j})}) - S_1(T_{b(\hat{\tau}_{1,j})})}{\sqrt{1 + M_1(T_{b(\hat{\tau}_{1,j})})}} > 4\sigma\sqrt{2\log(T)} + \sigma Q^{-1}(\frac{1}{1+\sqrt{x}})\right) \tag{69}$$

$$\leq \exp\left(-\frac{2\sigma^2(4\sqrt{2\log(T)} + Q^{-1}(\frac{1}{1+\sqrt{x}}))^2}{\alpha}\right) \tag{70}$$

$$\leq \exp\left(-\frac{16\sigma^2 \log(T)}{\alpha}\right)\exp\left(-\frac{2\sigma^2(2\sqrt{2\log(T)} + Q^{-1}(\frac{1}{1+\sqrt{x}}))^2}{\alpha}\right) \tag{71}$$

$$\leq \frac{1}{T} \tag{72}$$

(69) follows from (68). Lemma 6 and (66) lead to (70). Similarly (71) is due to (66) and the fact that $(a+b)^2 \geq a^2 + b^2$ for any non-negative $a$ and $b$. Finally, (72) follows from the fact that $\alpha \leq \frac{5\sigma^2}{4}$. However, for big $x$ values, we can provide a tighter upper bound. Firstly, $Q^{-1}(\frac{1}{1+\sqrt{x}}) \geq \sqrt{\frac{2}{3}\log(x)}$ if $x \geq x_0$ for some $x_0 \geq 4$ by (21). So for $x \geq x_0$

$$\exp\left(-\frac{2\sigma^2(2\sqrt{2\log(T)} + Q^{-1}(\frac{1}{1+\sqrt{x}}))^2}{\alpha}\right) \leq \exp(-\frac{4\sigma^2 \log(x)}{3\alpha}) \leq \frac{1}{x^{16/15}}$$

where we used the fact that $\alpha \leq \frac{5\sigma^2}{4}$. This inequality shows that

$$\mathbb{P}\left(\hat{\tau}_{1,j} < \infty, \frac{\sqrt{1 + M_1(T_{b(\hat{\tau}_{1,j})})}}{\sigma}\left(\frac{\mu_1 + \mu_i}{2} - \frac{S_1(T_{b(\hat{\tau}_{1,j})})}{1 + M_1(T_{b(\hat{\tau}_{1,j})})}\right) > Q^{-1}(\frac{1}{1+\sqrt{x}})\right) \leq \frac{1}{Tx^{16/15}} \tag{73}$$

if $x \geq x_0$ by (71). Overall, if we plug (72) and (73) back into (64), we have

$$\mathbb{E}\left[\mathbb{1}(\hat{\tau}_{1,j} < \infty)\left(\frac{1}{Q\left(\frac{\sqrt{1+M_1(T_{b(\hat{\tau}_{1,j})})}}{\sigma}\left(\frac{\mu_1+\mu_i}{2} - \frac{S_1(T_{b(\hat{\tau}_{1,j})})}{1+M_1(T_{b(\hat{\tau}_{1,j})})}\right)\right)} - 1\right)^2\right] \leq \frac{1+x_0}{T} + \int_{x=x_0}^{\infty} \frac{1}{Tx^{16/15}},$$

which finishes the proof since $x_0$ is an absolute constant. $\square$

## E    Proof of Theorem 1

First of all, (3) is the immediate result of (1) and (2). To prove (4), note that the regret contribution from the arms with $\Delta_i \leq \sigma\sqrt{\frac{\alpha K \log(T)}{T}}$ in the first $T$ rounds can not exceed $\sigma\sqrt{\alpha K T \log(T)}$.

As for any arm with $\Delta_i > \sigma\sqrt{\frac{\alpha K \log(T)}{T}}$, by (2) $\Delta_i \, \mathbb{E}[N_i(T)] \leq \frac{C_1 \sigma \sqrt{\alpha T \log(T)}}{\sqrt{K}}$, which leads to a maximum regret contribution of $C_1 \sigma \sqrt{\alpha K T \log(T)}$ from these arms. As a result, $\mathbb{E}[R(T)] \leq (1 + C_1)\sigma\sqrt{\alpha K T \log(T)}$ and this proves (4).

We will now prove (2). Lets pick any $i \geq 2$. We start the proof by first dividing $N_i(T)$ into smaller terms as follows:

$$\mathbb{E}[N_i(T)] = \mathbb{E}[\sum_{t=1}^{T} \mathbb{1}(A_t = i, \theta_i(t) > \frac{\mu_1 + \mu_i}{2})] + \mathbb{E}[\sum_{t=1}^{T} \mathbb{1}(A_t = i, \theta_i(t) \leq \frac{\mu_1 + \mu_i}{2})]$$

$$\leq \mathbb{E}[\sum_{t=1}^{T} \mathbb{1}(A_t = i, \theta_i(t) > \frac{\mu_1 + \mu_i}{2}, M_i(T_{b(t)}) \geq 32\sigma^2 \frac{\log(T)}{\Delta_i^2})] \tag{74}$$

$$+ \mathbb{E}[\sum_{t=1}^{T} \mathbb{1}(A_t = i, t = C_{e,k} \text{ for some } k, \theta_i(t) > \frac{\mu_1 + \mu_i}{2}, M_i(T_{b(t)}) < 32\sigma^2 \frac{\log(T)}{\Delta_i^2})] \tag{75}$$

$$+ \mathbb{E}[\sum_{t=1}^{T} \mathbb{1}(A_t = i, t \neq C_{e,k} \text{ for all } k, \theta_i(t) > \frac{\mu_1 + \mu_i}{2}, M_i(T_{b(t)}) < 32\sigma^2 \frac{\log(T)}{\Delta_i^2})] \tag{76}$$

$$+ 1 + \mathbb{E}[\sum_{t=2}^{T} \mathbb{1}(A_t = i, \theta_i(t) \leq \frac{\mu_1 + \mu_i}{2}, M_1(T_{b(t)}) \geq 32\sigma^2 \frac{\log(T)}{\Delta_i^2})] \tag{77}$$

$$+ \mathbb{E}[\sum_{t=2}^{T} \mathbb{1}(A_t = i, A_{t-1} = 1, \theta_i(t) \leq \frac{\mu_1 + \mu_i}{2}, M_1(T_{b(t)}) < 32\sigma^2 \frac{\log(T)}{\Delta_i^2})] \tag{78}$$

$$+ \mathbb{E}[\sum_{t=2}^{T} \mathbb{1}(A_t = i, A_{t-1} \neq 1, \theta_i(t) \leq \frac{\mu_1 + \mu_i}{2}, M_1(T_{b(t)}) < 32\sigma^2 \frac{\log(T)}{\Delta_i^2})] \tag{79}$$

Before we proceed to bound each of these terms, we provide our understanding as to how we decomposed this $N_i(T)$ term.

**How to Decompose $N_i(T)$ and Handle Each Term:** Since Thompson sampling relies on samples $\{\theta_i(t)\}_{i=1}^{K}$ to decide on the next action, it is natural that the term $N_i(T)$ is split whether $\theta_i(t)$ is above a certain threshold or not. In our analysis, we set this threshold to be $\frac{\mu_1 + \mu_i}{2}$ and the resulting terms are $\sum_{t=1}^{T} \mathbb{1}(A_t = i, \theta_i(t) > \frac{\mu_1 + \mu_i}{2})$ and $\sum_{t=1}^{T} \mathbb{1}(A_t = i, \theta_i(t) \leq \frac{\mu_1 + \mu_i}{2})$. We should note that as the agent observes more rewards from the $i^{th}$ arm, its samples $\theta_i(t)$ will concentrate around its true mean $\mu_i$ and the contribution from $\mathbb{1}(A_t = i, \theta_i(t) > \frac{\mu_1 + \mu_i}{2})$ terms will diminish. On the other hand, if the samples are already near their true mean, then as the agent receives more rewards from the optimal arm, her likelihood of playing the $i^{th}$ arm will decrease, i.e. the contributions from the terms $\mathbb{1}(A_t = i, \theta_i(t) \leq \frac{\mu_1 + \mu_i}{2})$ will diminish. Given this discussion, it is natural to center the decomposition of the terms $\sum_{t=1}^{T} \mathbb{1}(A_t = i, \theta_i(t) > \frac{\mu_1 + \mu_i}{2})$, i.e. (74)-(76), and $\sum_{t=1}^{T} \mathbb{1}(A_t = i, \theta_i(t) \leq \frac{\mu_1 + \mu_i}{2})$, i.e. (77)-(79), around $M_i$ and $M_1$ respectively. Earlier discussion also indicates that if there are enough observations from $i^{th}$ and the optimal arm, i.e. $M_i$ and $M_1$ are above reasonable thresholds, the agent's estimates should be close to the true values, which means that the terms in (74) and (77) should be small, i.e. of order $O(1)$. The main technical difficulty of analyzing the rest of the decomposition terms is that our algorithm uses random stopping times to decide on batch end points. So we need to show that these cycles are relatively short and their lengths do not diverge before the agent receives enough feedback. One way to show that these cycles are short is that we introduce conditions that naturally end the cycles, $\{t = C_{e,k}\}$ in (75) and $\{A_t = i, A_{t-1} = 1\}$ in (78). Since we assume that $M_i$ and $M_1$ are bounded in these terms, the contributions from (75) and (78) will be of order $O(\log(T))$. As for the terms in (76) and (79), even though it is not immediately clear that cycles containing these elements will be short, we can see that the conditions inside these indicator functions should naturally lead to short cycles. This is because in (76), even if the terms are not cycle end points and there not enough observations from the $i^{th}$ arm, $\{\theta_i(t) > \frac{\mu_1 + \mu_i}{2}\}$ condition can not persist for long. As for the term in (79), the conditions $\{A_{t-1} \neq 1\}$ and $\{\theta_i(t) \leq \frac{\mu_1 + \mu_i}{2}\}$ work against each other from the earlier discussion. As such, most of the proof is dedicated to showing that the terms in (76) and (79) indeed lead to short cycles

on average and they are of order $O(\log(T))$. We achieve this result by using a modified version of Lemma 2.8 [2] and the tails bounds specialized for our algorithm, which are the results from Section D. We now present the technical proof.

First of all, the terms with $M_i(T_{b(t)})$ or $M_1(T_{b(t)})$ being bigger than a certain level account for the estimation error and will be bounded by constants. We know that by (43) of Proposition 5 that

$$\mathbb{E}[\sum_{t=1}^{T} \mathbb{1}(A_t = i, \theta_i(t) > \frac{\mu_1 + \mu_i}{2}, M_i(T_{b(t)}) \geq 32\sigma^2 \frac{\log(T)}{\Delta_i^2})] \leq 2. \tag{80}$$

Now note that if $A_t = i$ and $\theta_i(t) \leq \frac{\mu_1 + \mu_i}{2}$, then $\theta_1(t) \leq \frac{\mu_1 + \mu_i}{2}$. That means

$$\mathbb{E}[\sum_{t=2}^{T} \mathbb{1}(A_t = i, \theta_i(t) \leq \frac{\mu_1 + \mu_i}{2}, M_1(T_{b(t)}) \geq 32\sigma^2 \frac{\log(T)}{\Delta_i^2})]$$

$$\leq \mathbb{E}[\sum_{t=2}^{T} \mathbb{1}(\theta_1(t) \leq \frac{\mu_1 + \mu_i}{2}, M_1(T_{b(t)}) \geq 32\sigma^2 \frac{\log(T)}{\Delta_i^2})] \leq 2 \tag{81}$$

where the last inequality follows from (42) of Proposition 5.

Now we bound the remaining terms. We first note that the condition $\{A_t = i, t = C_{e,k} \text{ for some } k\}$ signifies a cycle where the $i^{th}$ arm has been played at the end. Considering that the cycle count from the last batch can not increase more than its $\alpha$ multiple plus one by Algorithm 1, we know

$$\mathbb{E}[\sum_{t=1}^{T} \mathbb{1}(A_t = i, t = C_{e,k} \text{ for some } k, \theta_i(t) > \frac{\mu_1 + \mu_i}{2}, M_i(T_{b(t)}) < 32\sigma^2 \frac{\log(T)}{\Delta_i^2})] \leq 32\alpha\sigma^2 \frac{\log(T)}{\Delta_i^2} + 1 \tag{82}$$

due to the $\{M_i(T_{b(t)}) < 32\sigma^2 \frac{\log(T)}{\Delta_i^2}\}$ condition restricting the number of times we can count a unique cycle. Similarly the condition $\{A_t = i, A_{t-1} = 1\}$ means that $t-1$ is either a cycle beginning or end point, and again by $M_1(T_{b(t)}) < 32\sigma^2 \frac{\log(T)}{\Delta_i^2}$ limiting the number of times we can count a unique cycle with the first action we have

$$\mathbb{E}[\sum_{t=2}^{T} \mathbb{1}(A_t = i, A_{t-1} = 1, \theta_i(t) \leq \frac{\mu_1 + \mu_i}{2}, M_1(T_{b(t)}) < 32\sigma^2 \frac{\log(T)}{\Delta_i^2})] \leq 32\alpha\sigma^2 \frac{\log(T)}{\Delta_i^2} + 1. \tag{83}$$

Finally the only summands we did not bound are in (76) and (79). We will start with the harder (76) one and use the analysis there to bound (79) at the end. Let $\hat{\mathcal{H}}_t = \{\mathcal{H}_t, \theta_{A_1}(1), \theta_{A_2}(2), ..., \theta_{A_t}(t)\}$ and define the following stopping times for $\{\hat{\mathcal{H}}_t\}$

$$\tau_{b,k} = \begin{cases} \min\{t \in \mathbb{Z}^+ | A_t = i, t \neq C_{e,k} \text{ for all } k, \theta_i(t) > \frac{\mu_1 + \mu_i}{2}\} & \text{if } k = 1 \\ \min\{t \in \mathbb{Z}^+ | A_t = i, t \neq C_{e,k} \text{ for all } k, \theta_i(t) > \frac{\mu_1 + \mu_i}{2}, t > \tau_{e,k-1}\} & \text{if } k > 1 \end{cases}$$

and

$$\tau_{e,k} = \min\{t \in \mathbb{Z}^+ | A_t \neq i \text{ or } \theta_i(t) \leq \frac{\mu_1 + \mu_i}{2} \text{ such that } t > \tau_{b,k}\}.$$

Note that if any of these $\min$ operators are over an empty set, then the random variable is set to infinity. By the definitions of $\tau_{b,k}$ and $\tau_{e,k}$, it is easy to see that $\mathbb{1}(A_t = i, t \neq C_{e,k} \text{ for all } k, \theta_i(t) > \frac{\mu_1 + \mu_i}{2}, M_i(T_{b(t)}) < 32\sigma^2 \frac{\log(T)}{\Delta_i^2}) = 1$ only if $\tau_{b,k} \leq t < \tau_{e,k}$ for some $k$. This observations suggests that it is enough to only consider the intervals of $[\tau_{b,k}, \tau_{e,k} - 1]$ while summing over the elements of $1(A_t = i, t \neq C_{e,k} \text{ for all } k, \theta_i(t) > \frac{\mu_1 + \mu_i}{2}, M_i(T_{b(t)}) < 32\sigma^2 \frac{\log(T)}{\Delta_i^2})$ in (76). However, we can only consider an interval of $[\tau_{b,k}, \tau_{e,k} - 1]$ if $\tau_{b,k} < \infty$. That means we have the following bound

$$\sum_{t=1}^{T} \mathbb{1}(A_t = i, t \neq C_{e,k} \text{ for all } k, \theta_i(t) > \frac{\mu_1 + \mu_i}{2}, M_i(T_{b(t)}) < 32\sigma^2 \frac{\log(T)}{\Delta_i^2})$$

$$= \sum_{t=1}^{T} \mathbb{1}(A_t = i, t \neq C_{e,k} \text{ for all } k, \theta_i(t) > \frac{\mu_1 + \mu_i}{2}, M_i(T_{b(t)}) < 32\sigma^2 \frac{\log(T)}{\Delta_i^2}, M_1(t) \leq T)$$

(84)

$$\leq \sum_{k=1}^{\infty} \mathbb{1}(\tau_{b,k} < \infty)$$

$$\sum_{t=\tau_{b,k}}^{\tau_{e,k}-1} \mathbb{1}(A_t = i, t \neq C_{e,k} \text{ for all } k, \theta_i(t) > \frac{\mu_1 + \mu_i}{2}, M_i(T_{b(t)}) < 32\sigma^2 \frac{\log(T)}{\Delta_i^2}, M_1(t) \leq T)$$

(85)

$$\leq \sum_{k=1}^{\infty} \mathbb{1}(\tau_{b,k} < \infty, M_i(T_{b(\tau_{b,k})}) < 32\sigma^2 \frac{\log(T)}{\Delta_i^2}, M_1(\tau_{b,k}) \leq T)(\tau_{e,k} - \tau_{b,k}),$$

(86)

where (84) follows from the fact that $M_1(t) \leq T$ is satisfied for any $t \leq T$. Earlier discussion leads to (85). Finally, note that the time interval $[\tau_{b,k}, \tau_{e,k} - 1]$, where only the $i^{th}$ action is played, is inside a single cycle, so $M_i(T_{b(t)})$ and $M_1(t)$ stay the same for any $t \in [\tau_{b,k}, \tau_{e,k} - 1]$. This observation and ignoring the condition $\{A_t = i, t \neq C_{e,k} \text{ for all } k, \theta_i(t) > \frac{\mu_1 + \mu_i}{2}\}$ inside the indicator functions lead to (86). As the for expectation of the summands in (86):

$$\mathbb{E}[\mathbb{1}(\tau_{b,k} < \infty, M_i(T_{b(\tau_{b,k})}) < 32\sigma^2 \frac{\log(T)}{\Delta_i^2}, M_1(\tau_{b,k}) \leq T)(\tau_{e,k} - \tau_{b,k})]$$

$$= \mathbb{E}[\mathbb{E}[\mathbb{1}(\tau_{b,k} < \infty, M_i(T_{b(\tau_{b,k})}) < 32\sigma^2 \frac{\log(T)}{\Delta_i^2}, M_1(\tau_{b,k}) \leq T)(\tau_{e,k} - \tau_{b,k})|\hat{\mathcal{H}}_{\tau_{b,k}}]]$$

$$= \mathbb{E}[\mathbb{1}(\tau_{b,k} < \infty, M_i(T_{b(\tau_{b,k})}) < 32\sigma^2 \frac{\log(T)}{\Delta_i^2}, M_1(\tau_{b,k}) \leq T) \mathbb{E}[\tau_{e,k} - \tau_{b,k}|\hat{\mathcal{H}}_{\tau_{b,k}}]], \quad (87)$$

The last inequality is the result of the indicator function inside the expectation being measurable with respect to $\hat{\mathcal{H}}_{\tau_{b,k}}$. If $\tau_{b,k} < \infty$, then conditioned on $\hat{\mathcal{H}}_{\tau_{b,k}}$ we know that $\tau_{e,k} - \tau_{b,k} - 1$ is a geometric random variable for failures, where the the success probability is $1 - \mathbb{P}(A_{\tau_{b,k}+1} = i, \theta_i(\tau_{b,k} + 1) > \frac{\mu_1 + \mu_i}{2}|\hat{\mathcal{H}}_{\tau_{b,k}})$. The reason is that since $\tau_{b,k} < \infty$, we know that $\tau_{e,k} - \tau_{b,k} \geq 1$ and by the definitions of $\tau_{b,k}$ and $\tau_{e,k}$ $[\tau_{b,k}, \tau_{e,k}]$ defines a time interval in a single cycle like we mentioned earlier. These observations show that the sampling process remains the same throughout $[\tau_{b,k}, \tau_{e,k}]$, and conditioned on $\hat{\mathcal{H}}_{\tau_{b,k}}$ $[\tau_{b,k} + 1, \tau_{e,k} - 1]$ is a period of failures if we were to define success as $A_i(t) \neq i$ or $\theta_i(t) \leq \frac{\mu_1 + \mu_i}{2}$. The overall analysis proves the following set of inequalities

$$\mathbb{1}(\tau_{b,k} < \infty) \mathbb{E}[\tau_{e,k} - \tau_{b,k}|\hat{\mathcal{H}}_{\tau_{b,k}}] = \mathbb{1}(\tau_{b,k} < \infty) \frac{1}{1 - \mathbb{P}(A_{\tau_{b,k}+1} = i, \theta_i(\tau_{b,k} + 1) > \frac{\mu_1 + \mu_i}{2}|\hat{\mathcal{H}}_{\tau_{b,k}})}$$

$$\leq \mathbb{1}(\tau_{b,k} < \infty) \frac{1}{\mathbb{P}(\theta_i(\tau_{b,k} + 1) \leq \frac{\mu_1 + \mu_i}{2}|\hat{\mathcal{H}}_{\tau_{b,k}})}$$

almost surely. Putting this inequality inside (87) and summing the elements like in (86) shows that

$$\mathbb{E}[\sum_{t=1}^{T} \mathbb{1}(A_t = i, t \neq C_{e,k} \text{ for all } k, \theta_i(t) > \frac{\mu_1 + \mu_i}{2}, M_i(T_{b(t)}) < 32\sigma^2 \frac{\log(T)}{\Delta_i^2})]$$

$$\leq \mathbb{E}[\sum_{k=1}^{\infty} \mathbb{1}(\tau_{b,k} < \infty, M_i(T_{b(\tau_{b,k})}) < 32\sigma^2 \frac{\log(T)}{\Delta_i^2}, M_1(\tau_{b,k}) \leq T) \frac{1}{\mathbb{P}(\theta_i(\tau_{b,k} + 1) \leq \frac{\mu_1 + \mu_i}{2} | \hat{\mathcal{H}}_{\tau_{b,k}})}]$$

$$= \mathbb{E}[\sum_{k=1}^{\infty} \mathbb{1}(\tau_{b,k} < \infty, M_i(T_{b(\tau_{b,k})}) < 32\sigma^2 \frac{\log(T)}{\Delta_i^2}, M_1(\tau_{b,k}) \leq T, A_{\tau_{e,k}} = i) \frac{1}{\mathbb{P}(\theta_i(\tau_{b,k} + 1) \leq \frac{\mu_1 + \mu_i}{2} | \hat{\mathcal{H}}_{\tau_{b,k}})}] \tag{88}$$

$$+ \mathbb{E}[\sum_{k=1}^{\infty} \mathbb{1}(\tau_{b,k} < \infty, M_i(T_{b(\tau_{b,k})}) < 32\sigma^2 \frac{\log(T)}{\Delta_i^2}, M_1(\tau_{b,k}) \leq T, A_{\tau_{e,k}} \neq i) \frac{1}{\mathbb{P}(\theta_i(\tau_{b,k} + 1) \leq \frac{\mu_1 + \mu_i}{2} | \hat{\mathcal{H}}_{\tau_{b,k}})}] \tag{89}$$

Note by the earlier analysis we know that on $\{\tau_{b,k} < \infty\}$ $\tau_{e,k}$ is almost surely finite. Then the last equality follows from dividing the terms according to $A_{\tau_{e,k}} = i$ or not. We will first bound the summand in (88). To that end, we now analyze $\mathbb{1}(\tau_{b,k} < \infty) \mathbb{P}(A_{\tau_{e,k}} = 1 | \hat{\mathcal{H}}_{\tau_{b,k}})$. Note that by the earlier analysis we know the sampling distributions remains the same throughout $[\tau_{b,k}, \tau_{e,k}]$, which leads to

$$\mathbb{P}(A_{\tau_{e,k}} = 1 | \hat{\mathcal{H}}_{\tau_{b,k}}) = \frac{\mathbb{P}(A_{\tau_{b,k}+1} = 1 | \hat{\mathcal{H}}_{\tau_{b,k}})}{1 - \mathbb{P}(A_{\tau_{b,k}+1} = i, \theta_i(\tau_{b,k} + 1) > \frac{\mu_1 + \mu_i}{2} | \hat{\mathcal{H}}_{\tau_{b,k}})} \tag{90}$$

$$\geq \frac{\mathbb{P}(\theta_1(\tau_{b,k} + 1) > \frac{\mu_1 + \mu_i}{2}, \theta_j(\tau_{b,k} + 1) \leq \frac{\mu_1 + \mu_i}{2} \text{ for all } j \neq 1 | \hat{\mathcal{H}}_{\tau_{b,k}})}{1 - \mathbb{P}(A_{\tau_{b,k}+1} = i, \theta_i(\tau_{b,k} + 1) > \frac{\mu_1 + \mu_i}{2} | \hat{\mathcal{H}}_{\tau_{b,k}})} \tag{91}$$

$$= \frac{\mathbb{P}(\theta_1(\tau_{b,k} + 1) > \frac{\mu_1 + \mu_i}{2} | \hat{\mathcal{H}}_{\tau_{b,k}}) \mathbb{P}(\theta_j(\tau_{b,k} + 1) \leq \frac{\mu_1 + \mu_i}{2} \text{ for all } j \neq 1 | \hat{\mathcal{H}}_{\tau_{b,k}})}{1 - \mathbb{P}(A_{\tau_{b,k}+1} = i, \theta_i(\tau_{b,k} + 1) > \frac{\mu_1 + \mu_i}{2} | \hat{\mathcal{H}}_{\tau_{b,k}})} \tag{92}$$

on $\{\tau_{b,k} < \infty\}$. Note that since $\tau_{b,k} < \infty$, all the conditional probabilities stated here are almost surely positive. Here (90) trivially follows from the definition of success and failure of the geometric random variable we have defined earlier, i.e. $\tau_{e,k} - \tau_{b,k} - 1$. (91) is the result of the action selection process where we know that for the first action to be chosen the sample $\theta_1$ has to be at least as big as the other samples. Finally, conditioned on $\hat{\mathcal{H}}_{\tau_{b,k}}$, $\{\theta_j(\tau_{b,k} + 1)\}_{j=1}^{K}$ are independent, which results in (92). On the other hand

$$\mathbb{P}(A_{\tau_{e,k}} = i | \hat{\mathcal{H}}_{\tau_{b,k}}) = \frac{\mathbb{P}(A_{\tau_{b,k}+1} = i, \theta_i(\tau_{b,k} + 1) \leq \frac{\mu_1 + \mu_i}{2} | \hat{\mathcal{H}}_{\tau_{b,k}})}{1 - \mathbb{P}(A_{\tau_{b,k}+1} = i, \theta_i(\tau_{b,k} + 1) > \frac{\mu_1 + \mu_i}{2} | \hat{\mathcal{H}}_{\tau_{b,k}})} \tag{93}$$

$$\leq \frac{\mathbb{P}(\theta_j(\tau_{b,k} + 1) \leq \frac{\mu_1 + \mu_i}{2} \text{ for all } j | \hat{\mathcal{H}}_{\tau_{b,k}})}{1 - \mathbb{P}(A_{\tau_{b,k}+1} = i, \theta_i(\tau_{b,k} + 1) > \frac{\mu_1 + \mu_i}{2} | \hat{\mathcal{H}}_{\tau_{b,k}})} \tag{94}$$

$$= \frac{\mathbb{P}(\theta_1(\tau_{b,k} + 1) \leq \frac{\mu_1 + \mu_i}{2} | \hat{\mathcal{H}}_{\tau_{b,k}}) \mathbb{P}(\theta_j(\tau_{b,k} + 1) \leq \frac{\mu_1 + \mu_i}{2} \text{ for all } j \neq 1 | \hat{\mathcal{H}}_{\tau_{b,k}})}{1 - \mathbb{P}(A_{\tau_{b,k}+1} = i, \theta_i(\tau_{b,k} + 1) > \frac{\mu_1 + \mu_i}{2} | \hat{\mathcal{H}}_{\tau_{b,k}})} \tag{95}$$

on $\{\tau_{b,k} < \infty\}$. Note that (93) follows from the fact that $A_{\tau_{e,k}} = i$ only if $\theta_i(\tau_{e,k}) \leq \frac{\mu_1 + \mu_i}{2}$ by the definition of $\tau_{e,k}$. Considering that $\{A_{\tau_{b,k}+1} = i, \theta_i(\tau_{b,k} + 1) \leq \frac{\mu_1 + \mu_i}{2}\}$ means $\theta_j(\tau_{b,k} + 1) \leq \frac{\mu_1 + \mu_i}{2}$ for all $j$, we have (94). (95) is the result of the conditional independence. Combining (92) and (95) leads to

$$\mathbb{P}(A_{\tau_{e,k}} = i | \hat{\mathcal{H}}_{\tau_{b,k}}) \leq \frac{\mathbb{P}(\theta_1(\tau_{b,k} + 1) \leq \frac{\mu_1 + \mu_i}{2} | \hat{\mathcal{H}}_{\tau_{b,k}})}{\mathbb{P}(\theta_1(\tau_{b,k} + 1) > \frac{\mu_1 + \mu_i}{2} | \hat{\mathcal{H}}_{\tau_{b,k}})} \mathbb{P}(A_{\tau_{e,k}} = 1 | \hat{\mathcal{H}}_{\tau_{b,k}}). \tag{96}$$

on $\{\tau_{b,k} < \infty\}$. As a result

$$\mathbb{E}[\mathbb{1}(\tau_{b,k} < \infty, M_i(T_{b(\tau_{b,k})}) < 32\sigma^2 \frac{\log(T)}{\Delta_i^2}, M_1(\tau_{b,k}) \leq T, A_{\tau_{e,k}} = i) \frac{1}{\mathbb{P}(\theta_i(\tau_{b,k}+1) \leq \frac{\mu_1+\mu_i}{2}|\hat{\mathcal{H}}_{\tau_{b,k}})}]$$

$$= \mathbb{E}[\mathbb{E}[\mathbb{1}(\tau_{b,k} < \infty, M_i(T_{b(\tau_{b,k})}) < 32\sigma^2 \frac{\log(T)}{\Delta_i^2}, M_1(\tau_{b,k}) \leq T) \frac{1}{\mathbb{P}(\theta_i(\tau_{b,k}+1) \leq \frac{\mu_1+\mu_i}{2}|\hat{\mathcal{H}}_{\tau_{b,k}})}$$

$$\mathbb{1}(A_{\tau_{e,k}} = i)|\hat{\mathcal{H}}_{\tau_{b,k}}]]$$

$$= \mathbb{E}[\mathbb{1}(\tau_{b,k} < \infty, M_i(T_{b(\tau_{b,k})}) < 32\sigma^2 \frac{\log(T)}{\Delta_i^2}, M_1(\tau_{b,k}) \leq T) \frac{1}{\mathbb{P}(\theta_i(\tau_{b,k}+1) \leq \frac{\mu_1+\mu_i}{2}|\hat{\mathcal{H}}_{\tau_{b,k}})}$$

$$\mathbb{P}(A_{\tau_{e,k}} = i|\hat{\mathcal{H}}_{\tau_{b,k}})] \tag{97}$$

$$\leq \mathbb{E}[\mathbb{1}(\tau_{b,k} < \infty, M_i(T_{b(\tau_{b,k})}) < 32\sigma^2 \frac{\log(T)}{\Delta_i^2}, M_1(\tau_{b,k}) \leq T) \frac{1}{\mathbb{P}(\theta_i(\tau_{b,k}+1) \leq \frac{\mu_1+\mu_i}{2}|\hat{\mathcal{H}}_{\tau_{b,k}})}$$

$$\frac{\mathbb{P}(\theta_1(\tau_{b,k}+1) \leq \frac{\mu_1+\mu_i}{2}|\hat{\mathcal{H}}_{\tau_{b,k}})}{\mathbb{P}(\theta_1(\tau_{b,k}+1) > \frac{\mu_1+\mu_i}{2}|\hat{\mathcal{H}}_{\tau_{b,k}})} \mathbb{P}(A_{\tau_{e,k}} = 1|\hat{\mathcal{H}}_{\tau_{b,k}})] \tag{98}$$

$$= \mathbb{E}[\mathbb{1}(\tau_{b,k} < \infty, M_i(T_{b(\tau_{b,k})}) < 32\sigma^2 \frac{\log(T)}{\Delta_i^2}, M_1(\tau_{b,k}) \leq T) \frac{1}{\mathbb{P}(\theta_i(\tau_{b,k}+1) \leq \frac{\mu_1+\mu_i}{2}|\hat{\mathcal{H}}_{\tau_{b,k}})}$$

$$\frac{\mathbb{P}(\theta_1(\tau_{b,k}+1) \leq \frac{\mu_1+\mu_i}{2}|\hat{\mathcal{H}}_{\tau_{b,k}})}{\mathbb{P}(\theta_1(\tau_{b,k}+1) > \frac{\mu_1+\mu_i}{2}|\hat{\mathcal{H}}_{\tau_{b,k}})} \mathbb{1}(A_{\tau_{e,k}} = 1)]$$

$$\leq \mathbb{E}[\mathbb{1}(\tau_{b,k} < \infty, A_{\tau_{e,k}} = 1, M_i(T_{b(\tau_{b,k})}) < 32\sigma^2 \frac{\log(T)}{\Delta_i^2}, M_1(\tau_{b,k}) \leq T) \frac{1}{\mathbb{P}^2(\theta_i(\tau_{b,k}+1) \leq \frac{\mu_1+\mu_i}{2}|\hat{\mathcal{H}}_{\tau_{b,k}})}]$$

$$+ \mathbb{E}[\mathbb{1}(\tau_{b,k} < \infty, A_{\tau_{e,k}} = 1, M_i(T_{b(\tau_{b,k})}) < 32\sigma^2 \frac{\log(T)}{\Delta_i^2}, M_1(\tau_{b,k}) \leq T) \frac{\mathbb{P}^2(\theta_1(\tau_{b,k}+1) \leq \frac{\mu_1+\mu_i}{2}|\hat{\mathcal{H}}_{\tau_{b,k}})}{\mathbb{P}^2(\theta_1(\tau_{b,k}+1) > \frac{\mu_1+\mu_i}{2}|\hat{\mathcal{H}}_{\tau_{b,k}})}].$$
$$\tag{99}$$

(97) follows from the measurability of the indicator function and the inverse probability term with respect to $\hat{\mathcal{H}}_{\tau_{b,k}}$. (96) leads to (98). Finally, the last inequality follows from the fact that $2\sqrt{a \times b} \leq a + b$ for any non-negative $a$ and $b$. Here (99) shows that

$$\mathbb{E}[\sum_{k=1}^{\infty} \mathbb{1}(\tau_{b,k} < \infty, M_i(T_{b(\tau_{b,k})}) < 32\sigma^2 \frac{\log(T)}{\Delta_i^2}, M_1(\tau_{b,k}) \leq T, A_{\tau_{e,k}} = i) \frac{1}{\mathbb{P}(\theta_i(\tau_{b,k}+1) \leq \frac{\mu_1+\mu_i}{2}|\hat{\mathcal{H}}_{\tau_{b,k}})}]$$

$$\leq \mathbb{E}[\sum_{k=1}^{\infty} \mathbb{1}(\tau_{b,k} < \infty, A_{\tau_{e,k}} = 1, M_i(T_{b(\tau_{b,k})}) < 32\sigma^2 \frac{\log(T)}{\Delta_i^2}) \frac{1}{\mathbb{P}^2(\theta_i(\tau_{b,k}+1) \leq \frac{\mu_1+\mu_i}{2}|\hat{\mathcal{H}}_{\tau_{b,k}})}]$$

$$+ \mathbb{E}[\sum_{k=1}^{\infty} \mathbb{1}(\tau_{b,k} < \infty, A_{\tau_{e,k}} = 1, M_1(\tau_{b,k}) \leq T) \frac{\mathbb{P}^2(\theta_1(\tau_{b,k}+1) \leq \frac{\mu_1+\mu_i}{2}|\hat{\mathcal{H}}_{\tau_{b,k}})}{\mathbb{P}^2(\theta_1(\tau_{b,k}+1) > \frac{\mu_1+\mu_i}{2}|\hat{\mathcal{H}}_{\tau_{b,k}})}] \tag{100}$$

Here we eliminated one condition from each indicator function in the last inequality. However, we know by the action selection process of Thompson sampling and $b(\tau_{b,k}+1) = b(\tau_{b,k})$ equality due to $\tau_{b,k}$ and $\tau_{b,k} + 1$ being in the same cycle that

$$\mathbb{P}(\theta_1(\tau_{b,k}+1) > \frac{\mu_1+\mu_i}{2}|\hat{\mathcal{H}}_{\tau_{b,k}}) = Q\left(\frac{\sqrt{1 + M_1(T_{b(\tau_{b,k})})}}{\sigma}\left(\frac{\mu_1+\mu_i}{2} - \frac{S_1(T_{b(\tau_{b,k})})}{1 + M_1(T_{b(\tau_{b,k})})}\right)\right) \tag{101}$$

and

$$\mathbb{P}(\theta_i(\tau_{b,k}+1) \leq \frac{\mu_1+\mu_i}{2}|\hat{\mathcal{H}}_{\tau_{b,k}}) = Q\left(\frac{\sqrt{1 + M_i(T_{b(\tau_{b,k})})}}{\sigma}\left(\frac{S_i(T_{b(\tau_{b,k})})}{1 + M_i(T_{b(\tau_{b,k})})} - \frac{\mu_1+\mu_i}{2}\right)\right) \tag{102}$$

on $\{\tau_{b,k} < \infty\}$. Considering (101) and (102), we see can view (100) as

$$\mathbb{E}[\sum_{k=1}^{\infty} \mathbb{1}(\tau_{b,k} < \infty, M_i(T_{b(\tau_{b,k})}) < 32\sigma^2 \frac{\log(T)}{\Delta_i^2}, M_1(\tau_{b,k}) \leq T, A_{\tau_{e,k}} = i) \frac{1}{\mathbb{P}(\theta_i(\tau_{b,k}+1) \leq \frac{\mu_1+\mu_i}{2}|\hat{\mathcal{H}}_{\tau_{b,k}})}]$$

$$\leq \mathbb{E}[\sum_{k=1}^{\infty} \mathbb{1}(\tau_{b,k} < \infty, A_{\tau_{e,k}} = 1, M_i(T_{b(\tau_{b,k})}) < 32\sigma^2 \frac{\log(T)}{\Delta_i^2}) f_1(M_i(T_{b(\tau_{b,k})}), S_i(T_{b(\tau_{b,k})}))]$$

$$+ \mathbb{E}[\sum_{k=1}^{\infty} \mathbb{1}(\tau_{b,k} < \infty, A_{\tau_{e,k}} = 1, M_1(\tau_{b,k}) \leq T) f_2(M_1(T_{b(\tau_{b,k})}), S_1(T_{b(\tau_{b,k})}))]$$

where $f_1$ and $f_2$ are some functions with domain $\mathbb{R}_{\geq 0} \times \mathbb{R}$. If we let

$$\hat{\tau}_{i,j} = \begin{cases} \min\{t \in \mathbb{Z}^+ | A_t = i, t = C_{b,k} \text{ or } C_{e,k} \text{ for some } k\} & \text{if } j = 1 \\ \min\{t \in \mathbb{Z}^+ | A_t = i, t = C_{b,k} \text{ or } C_{e,k} \text{ for some } k, t > \hat{\tau}_{i,j-1}\} & \text{if } j > 1, \end{cases}$$

which denotes the time indices where we choose the $i^{th}$ arm at the beginning or at the end of a cycle, we notice that $M_i(T_{b(\tau_{b,k})}) = M_i(T_{b(\hat{\tau}_{i,j})})$ and $S_i(T_{b(\tau_{b,k})}) = S_i(T_{b(\hat{\tau}_{i,j})})$ for some $j$ on $\{\tau_{b,k} < \infty\}$ since $\tau_{b,k} < \infty$ means that the agent has played the $i^{th}$ action at the beginning of the cycle containing $\tau_{b,k}$. However, when we look $\mathbb{1}(\tau_{b,k} < \infty, A_{\tau_{e,k}} = 1, M_i(T_{b(\tau_{b,k})}) < 32\sigma^2 \frac{\log(T)}{\Delta_i^2}) f_1(M_i(T_{b(\tau_{b,k})}), S_i(T_{b(\tau_{b,k})}))$ terms, we realize that each time interval $[\tau_{b,k}, \tau_{e,k} - 1]$ will belong to a different cycle due to $A_{\tau_{e,k}} = 1$, and the condition $\{M_i(T_{b(\tau_{b,k})}) < 32\sigma^2 \frac{\log(T)}{\Delta_i^2}\}$ implies that the indicator function can be non-zero only if $[\tau_{b,k}, \tau_{e,k}]$ is inside the one of the first $\lceil 32\alpha\sigma^2 \frac{\log(T)}{\Delta_i^2} \rceil$ cycles of the $i^{th}$ arm. The overall discussion leads to the following bound:

$$\sum_{k=1}^{\infty} \mathbb{1}(\tau_{b,k} < \infty, A_{\tau_{e,k}} = 1, M_i(T_{b(\tau_{b,k})}) < 32\sigma^2 \frac{\log(T)}{\Delta_i^2}) f_1(M_i(T_{b(\tau_{b,k})}), S_i(T_{b(\tau_{b,k})}))$$

$$\leq \sum_{j=1}^{\lceil 32\alpha\sigma^2 \frac{\log(T)}{\Delta_i^2} \rceil} \mathbb{1}(\hat{\tau}_{i,j} < \infty) f_1(M_i(T_{b(\hat{\tau}_{i,j})}), S_i(T_{b(\hat{\tau}_{i,j})})). \tag{103}$$

On the other hand, on $\{\tau_{b,k} < \infty, A_{\tau_{e,k}} = 1\}$, $M_1(T_{b(\tau_{b,k})}) = M_1(T_{b(\hat{\tau}_{1,j})})$ and $S_1(T_{b(\tau_{b,k})}) = S_1(T_{b(\hat{\tau}_{1,j})})$ for some $j$ since $\tau_{e,k}$ here is the cycle end point. Similar to the earlier analysis, for each $k$ that satisfies the $\{\tau_{b,k} < \infty, A_{\tau_{e,k}} = 1, M_1(\tau_{b,k}) \leq T\}$ condition, $[\tau_{b,k}, \tau_{e,k}]$ will be in a distinct cycle from the first $T+1$ ones containing the first arm. This observation shows that

$$\sum_{k=1}^{\infty} \mathbb{1}(\tau_{b,k} < \infty, A_{\tau_{e,k}} = 1, M_1(\tau_{b,k}) \leq T) f_2(M_1(T_{b(\tau_{b,k})}), S_1(T_{b(\tau_{b,k})}))$$

$$\leq \sum_{j=1}^{T+1} \mathbb{1}(\hat{\tau}_{1,j} < \infty) f_2(M_1(T_{b(\hat{\tau}_{1,j})}), S_1(T_{b(\hat{\tau}_{1,j})})) \tag{104}$$

In view of (100), (103) and (104) result in the following bound

$$\mathbb{E}[\sum_{k=1}^{\infty} \mathbb{1}(\tau_{b,k} < \infty, M_i(T_{b(\tau_{b,k})}) < 32\sigma^2 \frac{\log(T)}{\Delta_i^2}, M_1(\tau_{b,k}) \leq T, A_{\tau_{e,k}} = i) \frac{1}{\mathbb{P}(\theta_i(\tau_{b,k}+1) \leq \frac{\mu_1+\mu_i}{2}|\hat{\mathcal{H}}_{\tau_{b,k}})}]$$

$$\leq \mathbb{E}\left[ \sum_{j=1}^{\lceil 32\alpha\sigma^2 \frac{\log(T)}{\Delta_i^2} \rceil} \mathbb{1}(\hat{\tau}_{i,j} < \infty) \frac{1}{Q^2\left( \frac{\sqrt{1+M_i(T_{b(\hat{\tau}_{i,j})})}}{\sigma} \left( \frac{S_i(T_{b(\hat{\tau}_{i,j})})}{1+M_i(T_{b(\hat{\tau}_{i,j})})} - \frac{\mu_1+\mu_i}{2} \right) \right)} \right]$$

$$+ \mathbb{E}\left[ \sum_{j=1}^{T+1} \mathbb{1}(\hat{\tau}_{1,j} < \infty) \left( \frac{1}{Q\left( \frac{\sqrt{1+M_1(T_{b(\hat{\tau}_{1,j})})}}{\sigma} \left( \frac{\mu_1+\mu_i}{2} - \frac{S_1(T_{b(\hat{\tau}_{1,j})})}{1+M_1(T_{b(\hat{\tau}_{1,j})})} \right) \right)} - 1 \right)^2 \right]. \tag{105}$$

where we replaced $f_1$ and $f_2$ with their exact forms. Note that although we did not define $f_1$ and $f_2$ explicitly, it is easy understand their exact formulation from the earlier discussion, i.e. from the conditional probability functions in (100) and the equalities stated in (101) and (102). Here we know that the first expectation to the right-side of the inequality is upper bounded by $C(1 + 32\alpha\sigma^2\frac{\log(T)}{\Delta_i^2})$ by Proposition 7, where $C$ is an absolute constant. On the other hand, we have

$$\mathbb{E}\left[\mathbb{1}(\hat{\tau}_{1,j} < \infty)\left(\frac{1}{Q\left(\frac{\sqrt{1+M_1(T_{b(\hat{\tau}_{1,j})})}}{\sigma}\left(\frac{\mu_1+\mu_i}{2} - \frac{S_1(T_{b(\hat{\tau}_{1,j})})}{1+M_1(T_{b(\hat{\tau}_{1,j})})}\right)\right)} - 1\right)^2\right]$$

$$\leq \begin{cases} C & \text{if } j \leq 1024\alpha\sigma^2\frac{\log(T)}{\Delta_i^2} \\ C/T & \text{if } j > 1024\alpha\sigma^2\frac{\log(T)}{\Delta_i^2}, \end{cases}$$

where $C$ is an absolute constant independent of the system variables. Note that the constant bound follows from Proposition 7, while $O(1/T)$ bound is the result of Proposition 8. This overall analysis and (105) lead to

$$\mathbb{E}[\sum_{k=1}^{\infty}\mathbb{1}(\tau_{b,k} < \infty, M_i(T_{b(\tau_{b,k})}) < 32\sigma^2\frac{\log(T)}{\Delta_i^2}, M_1(\tau_{b,k}) \leq T, A_{\tau_{e,k}} = i)\frac{1}{\mathbb{P}(\theta_i(\tau_{b,k}+1) \leq \frac{\mu_1+\mu_i}{2}|\hat{\mathcal{H}}_{\tau_{b,k}})}]$$

$$\leq C(2 + 1056\alpha\sigma^2\frac{\log(T)}{\Delta_i^2}) \tag{106}$$

for an absolute constant $C$. This proof bounds the term in (88). However, with the analysis we have done so far, bounding the term in (89) is almost immediate. First note that, similar to the earlier analysis, $M_i(T_{b(\tau_{b,k})}) = M_i(T_{b(\hat{\tau}_{i,j})})$ and $S_i(T_{b(\tau_{b,k})}) = S_i(T_{b(\hat{\tau}_{i,j})})$ for some $j$ on $\{\tau_{b,k} < \infty\}$ since $\tau_{b,k} < \infty$ means that the agent has played the $i^{th}$ action at the beginning of the cycle containing $\tau_{b,k}$. Then by (102)

$$\mathbb{P}(\theta_i(\tau_{b,k}+1) \leq \frac{\mu_1+\mu_i}{2}|\hat{\mathcal{H}}_{\tau_{b,k}}) = Q\left(\frac{\sqrt{1+M_i(T_{b(\hat{\tau}_{i,j})})}}{\sigma}\left(\frac{S_i(T_{b(\hat{\tau}_{i,j})})}{1+M_i(T_{b(\hat{\tau}_{i,j})})} - \frac{\mu_1+\mu_i}{2}\right)\right)$$

on $\{\tau_{b,k} < \infty\}$ for some $j$. However, for each $k$ that satisfies the condition $\{\tau_{b,k} < \infty, M_i(T_{b(\tau_{b,k})}) < 32\sigma^2\frac{\log(T)}{\Delta_i^2}, M_1(\tau_{b,k}) \leq T, A_{\tau_{e,k}} \neq i\}$, $[\tau_{b,k}, \tau_{e,k}]$ has to be in a distinct cycle from the first $\lceil 32\alpha\sigma^2\frac{\log(T)}{\Delta_i^2}\rceil$ cycles of the $i^{th}$ arm. Note that the distinctiveness follows from the fact that $A_{\tau_{e,k}} \neq i$ condition ends the cycle, while the upper bound on the number of cycles is the result of $\{M_i(T_{b(\tau_{b,k})}) < 32\sigma^2\frac{\log(T)}{\Delta_i^2}\}$ and the way Algorithm 1 is implemented. These arguments naturally lead to

$$\mathbb{E}[\sum_{k=1}^{\infty}\mathbb{1}(\tau_{b,k} < \infty, M_i(T_{b(\tau_{b,k})}) < 32\sigma^2\frac{\log(T)}{\Delta_i^2}, M_1(\tau_{b,k}) \leq T, A_{\tau_{e,k}} \neq i)\frac{1}{\mathbb{P}(\theta_i(\tau_{b,k}+1) \leq \frac{\mu_1+\mu_i}{2}|\hat{\mathcal{H}}_{\tau_{b,k}})}]$$

$$\leq \mathbb{E}\left[\sum_{j=1}^{\lceil 32\alpha\sigma^2\frac{\log(T)}{\Delta_i^2}\rceil}\mathbb{1}(\hat{\tau}_{i,j} < \infty)\frac{1}{Q\left(\frac{\sqrt{1+M_i(T_{b(\hat{\tau}_{i,j})})}}{\sigma}\left(\frac{S_i(T_{b(\hat{\tau}_{i,j})})}{1+M_i(T_{b(\hat{\tau}_{i,j})})} - \frac{\mu_1+\mu_i}{2}\right)\right)}\right]$$

$$\leq C(1 + 32\alpha\sigma^2\frac{\log(T)}{\Delta_i^2}) \tag{107}$$

where the last inequality follows from Proposition 7 and the range of $Q$ being from zero to one. In view of (88) and (89), combining (106) and (107) shows that

$$\mathbb{E}[\sum_{t=1}^{T}\mathbb{1}(A_t = i, t \neq C_{e,k} \text{ for all } k, \theta_i(t) > \frac{\mu_1+\mu_i}{2}, M_i(T_{b(t)}) < 32\sigma^2\frac{\log(T)}{\Delta_i^2})]$$

$$\leq C(3 + 1088\alpha\sigma^2\frac{\log(T)}{\Delta_i^2}) \tag{108}$$

where $C$ is an absolute constant. This finishes the analysis of the summand in (76).

Finally, we will bound the summand in (79). However, most of the proof ideas will follow from earlier analysis. First note that if $\theta_1(t) > \frac{\mu_1 + \mu_i}{2}$, while $\theta_j(t) \leq \frac{\mu_1 + \mu_i}{2}$ for $j \geq 2$, then $A_t = 1$:

$$\mathbb{P}(A_t = 1 \mid \mathcal{H}_{t-1}) \geq \mathbb{P}(\theta_1(t) > \frac{\mu_1 + \mu_i}{2} \mid \mathcal{H}_{t-1}) \, \mathbb{P}(\theta_j(t) \leq \frac{\mu_1 + \mu_i}{2} \text{ for all } j \neq 1 \mid \mathcal{H}_{t-1}) \quad (109)$$

where we also used the conditional independence of $\theta_j(t)$s given $\mathcal{H}_{t-1}$. On the other hand, if $A_t = i$ and $\theta_i(t) \leq \frac{\mu_1 + \mu_i}{2}$, then $\theta_j(t) \leq \frac{\mu_1 + \mu_i}{2}$ for all $j \geq 2$:

$$\mathbb{P}(A_t = i, \theta_i(t) \leq \frac{\mu_1 + \mu_i}{2} \mid \mathcal{H}_{t-1}) \leq \mathbb{P}(\theta_j(t) \leq \frac{\mu_1 + \mu_i}{2} \text{ for all } j \neq 1 \mid \mathcal{H}_{t-1}). \quad (110)$$

The combination of (109) and (110) lead to

$$\mathbb{P}(A_t = i, \theta_i(t) \leq \frac{\mu_1 + \mu_i}{2} \mid \mathcal{H}_{t-1}) \leq \frac{\mathbb{P}(A_t = 1 \mid \mathcal{H}_{t-1})}{\mathbb{P}(\theta_1(t) > \frac{\mu_1 + \mu_i}{2} \mid \mathcal{H}_{t-1})}. \quad (111)$$

Note that considering the action selection process of Algorithm 1 where conditioned on the past observations $\theta_1$ has a Gaussian distribution, $\mathbb{P}(\theta_1(t) > \frac{\mu_1 + \mu_i}{2} \mid \mathcal{H}_{t-1})$ will almost surely be non-zero. Then we have

$$\mathbb{E}[\mathbb{1}(A_t = i, A_{t-1} \neq 1, \theta_i(t) \leq \frac{\mu_1 + \mu_i}{2}, M_1(T_{b(t)}) < 32\sigma^2 \frac{\log(T)}{\Delta_i^2})]$$

$$= \mathbb{E}[\mathbb{E}[\mathbb{1}(A_t = i, A_{t-1} \neq 1, \theta_i(t) \leq \frac{\mu_1 + \mu_i}{2}, M_1(T_{b(t)}) < 32\sigma^2 \frac{\log(T)}{\Delta_i^2}) \mid \mathcal{H}_{t-1}]]$$

$$= \mathbb{E}[\mathbb{P}(A_t = i, \theta_i(t) \leq \frac{\mu_1 + \mu_i}{2} \mid \mathcal{H}_{t-1}) \, \mathbb{1}(A_{t-1} \neq 1, M_1(T_{b(t)}) < 32\sigma^2 \frac{\log(T)}{\Delta_i^2})] \quad (112)$$

$$\leq \mathbb{E}[\frac{\mathbb{P}(A_t = 1 \mid \mathcal{H}_{t-1})}{\mathbb{P}(\theta_1(t) > \frac{\mu_1 + \mu_i}{2} \mid \mathcal{H}_{t-1})} \, \mathbb{1}(A_{t-1} \neq 1, M_1(T_{b(t)}) < 32\sigma^2 \frac{\log(T)}{\Delta_i^2})] \quad (113)$$

$$= \mathbb{E}[\frac{1}{\mathbb{P}(\theta_1(t) > \frac{\mu_1 + \mu_i}{2} \mid \mathcal{H}_{t-1})} \, \mathbb{1}(A_t = 1, A_{t-1} \neq 1, M_1(T_{b(t)}) < 32\sigma^2 \frac{\log(T)}{\Delta_i^2})], \quad (114)$$

where (112) follows from moving terms measurable with respect to $\mathcal{H}_{t-1}$ out of the conditional expectation. The bound in (111) leads to (113). Finally, using (114) in (79) shows that

$$\mathbb{E}[\sum_{t=2}^{T} \mathbb{1}(A_t = i, A_{t-1} \neq 1, \theta_i(t) \leq \frac{\mu_1 + \mu_i}{2}, M_1(T_{b(t)}) < 32\sigma^2 \frac{\log(T)}{\Delta_i^2})]$$

$$\leq \mathbb{E}[\sum_{t=2}^{T} \frac{1}{\mathbb{P}(\theta_1(t) > \frac{\mu_1 + \mu_i}{2} \mid \mathcal{H}_{t-1})} \, \mathbb{1}(A_t = 1, A_{t-1} \neq 1, M_1(T_{b(t)}) < 32\sigma^2 \frac{\log(T)}{\Delta_i^2})] \quad (115)$$

Note here that if $A_t = 1$ and $A_{t-1} \neq 1$, then $t$ is either a cycle beginning or a cycle end point, which means that $t = \hat{\tau}_{1,j}$ for some $j$:

$$\mathbb{P}(\theta_1(t) > \frac{\mu_1 + \mu_i}{2} \mid \mathcal{H}_{t-1}) = Q\left( \frac{\sqrt{1 + M_1(T_{b(t)})}}{\sigma} \left( \frac{\mu_1 + \mu_i}{2} - \frac{S_1(T_{b(t)})}{1 + M_1(T_{b(t)})} \right) \right) \quad (116)$$

$$= Q\left( \frac{\sqrt{1 + M_1(T_{b(\hat{\tau}_{1,j})})}}{\sigma} \left( \frac{\mu_1 + \mu_i}{2} - \frac{S_1(T_{b(\hat{\tau}_{1,j})})}{1 + M_1(T_{b(\hat{\tau}_{1,j})})} \right) \right) \quad (117)$$

where the fact that conditioned on $\mathcal{H}_{t-1}$, $\theta_1(t) \sim \mathcal{N}(\frac{S_1(T_{b(t)})}{1+M_1(T_{b(t)})}, \frac{\sigma^2}{1+M_1(T_{b(t)})})$ leads to (116). In addition, each $t$ that satisfies $\{A_t = 1, A_{t-1} \neq 1, M_1(T_{b(t)}) < 32\sigma^2 \frac{\log(T)}{\Delta_i^2}\}$ condition has to belong to a different cycle and the index $j$ can not be bigger than $\lceil 32\alpha\sigma^2 \frac{\log(T)}{\Delta_i^2} \rceil$. So, in view of (115), (117)

leads to

$$\mathbb{E}[\sum_{t=2}^{T} \mathbb{1}(A_t = i, A_{t-1} \neq 1, \theta_i(t) \leq \frac{\mu_1 + \mu_i}{2}, M_1(T_{b(t)}) < 32\sigma^2 \frac{\log(T)}{\Delta_i^2})]$$

$$\leq \mathbb{E}\left[ \sum_{j=1}^{\lceil 32\alpha\sigma^2 \frac{\log(T)}{\Delta_i^2} \rceil} \mathbb{1}(\hat{\tau}_{1,j} < \infty) \frac{1}{Q\left( \frac{\sqrt{1+M_1(T_{b(\hat{\tau}_{1,j})})}}{\sigma} \left( \frac{\mu_1+\mu_i}{2} - \frac{S_1(T_{b(\hat{\tau}_{1,j})})}{1+M_1(T_{b(\hat{\tau}_{1,j})})} \right) \right)} \right]$$

$$\leq C(1 + 32\alpha\sigma^2 \frac{\log(T)}{\Delta_i^2}). \tag{118}$$

Here the last inequality is the application of Proposition 7. However, this result finishes the proof of (2) since the collection of bounds, (80), (81), (82), (83), (108), (118), prove that

$$\mathbb{E}[N_i(T)] \leq 6 + 64\alpha\sigma^2 \frac{\log(T)}{\Delta_i^2} + C(4 + 1120\alpha\sigma^2 \frac{\log(T)}{\Delta_i^2}).$$

# F  Additional Experiments

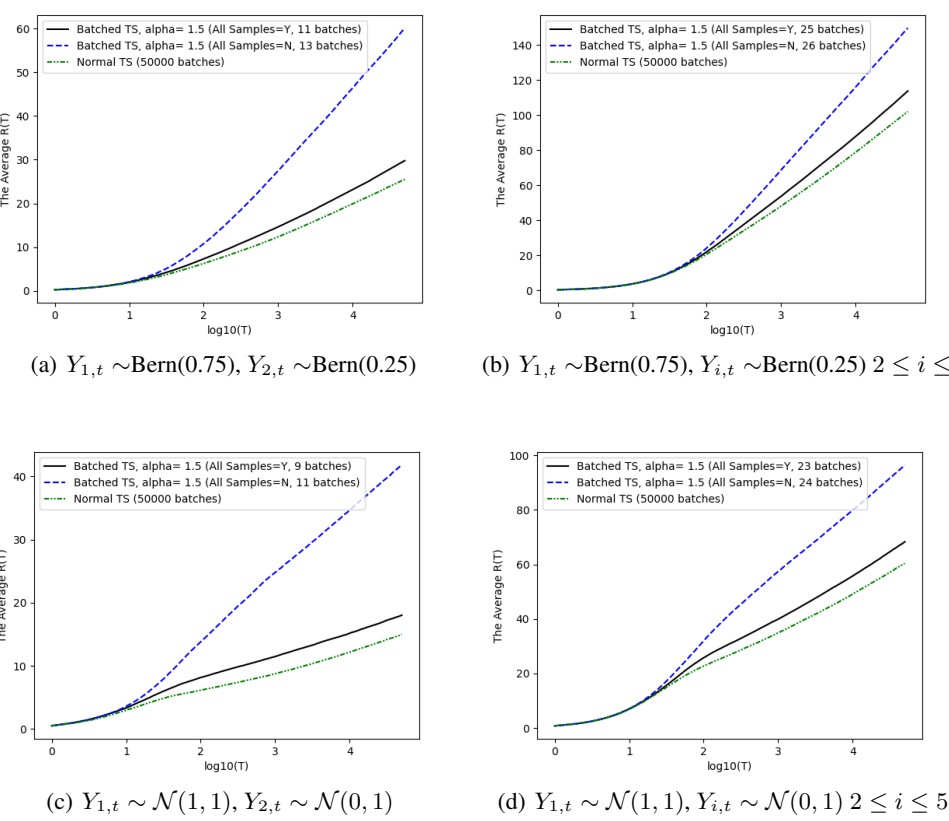

(a) $Y_{1,t} \sim \text{Bern}(0.75), Y_{2,t} \sim \text{Bern}(0.25)$

(b) $Y_{1,t} \sim \text{Bern}(0.75), Y_{i,t} \sim \text{Bern}(0.25)$ $2 \leq i \leq 5$

(c) $Y_{1,t} \sim \mathcal{N}(1,1), Y_{2,t} \sim \mathcal{N}(0,1)$

(d) $Y_{1,t} \sim \mathcal{N}(1,1), Y_{i,t} \sim \mathcal{N}(0,1)$ $2 \leq i \leq 5$

Figure 2: Empirical Regret Performance of Batched and normal Thompson sampling

In addition to the figures in Section 5, here we provide experiments that showcase how Algorithm 1 that does not make use of all reward observations (shown as All Samples=N) performs compared to the version that uses all the samples (shown as All Samples=Y). Additionally, instead of the average batch count, we report the $95\%$ percentile batch complexity required for each algorithm. As can be seen from these figures, the version of Batched Thompson sampling using all the samples

significantly outperforms the version that does not make use of all the observations. On the other hand, we note that the algorithm that does skip samples (All Samples=N) may require exponentially fewer observations on average than the other version (All Samples=Y). For example, in Figure 2 (a), Algorithm 1 (All Samples=N) uses around 400 samples on average for each of the two arms, while the batched TS algorithm that uses all the samples (All Samples=Y) uses $5 \times 10^4$ samples in total. Overall, this indicates a trade-off between fewer samples and possibly lower computation time versus better empirical regret performance.