# OpenReview forum: "Batched Thompson Sampling"
_NeurIPS.cc/2021/Conference — NeurIPS 2021 Poster_

### Official Review · Reviewer_7wHj · 2021-06-30

**Rating:** 7
**Confidence:** 3

**Summary:**

The paper addresses the problem of K-armed bandits in a batched setting. In this setting, the entire time duration is split into small batches and the feedback is received only at the end of each batch. The authors design a batched version of Thompson sampling, where the batch duration is set adaptively based on the past data. The author analyzes the proposed algorithm under Gaussian reward distribution and provides a problem-dependent O(log(T)) regret bound, as well as O(\sqrt{T}) minimax regret bound. The number of batches used is shown to be O(log(T)) in the worst case and admits O(log log(T)) amortized problem-dependent bound.  The authors validate their theoretical results using synthetic experiments.

**Limitations And Societal Impact:**

The paper is theoretical in nature, and the scope for negative societal impact seems very limited.

The authors may highlight the limitation of adaptivity in designing the batch size in practice.

**Main Review:**

Pros:
 - The proposed algorithm that adapts the batch size based on how many times the arms are played in the past is novel. They introduce a notion of the cycle -- a series of rounds with exactly two distinct arms. A batch ends when, for any arm, the number of cycles that arm is played in the batch is \alpha times the number of cycles that arm is played before the batch.

- The regret analysis builds on the proposed adaptive batching. It also uses the fact that the prior for arm rewards used in Thompson sampling remains unchanged during a batch. The amortized problem-dependent batching analysis is also novel.

- The writing is clear although the use of notations can be a little overwhelming at times.

- The paper mentions the relevant works to the best of my knowledge.


Drawbacks and Suggestions:
- A discussion on the tradeoffs of setting alpha close to 1 or away from 1 will be useful.

- A discussion on how the regret analysis with K=2 will be extended to K > 2 in the full proof will be insightful. With the cycle structure, the transition from K=2 to K > 2 seems non-trivial.

- The experiments are not run for long enough time, as evident from the linear regret shape. I suggest running the algorithm for enough time to obtain the log(T) shaped curve and use logscale for Y-axis.

- In Theorem 1 and Theorem 2 please specify that \alpha > 1.


**Time Spent Reviewing:**

6

---

> ### Author Response · Authors · 2021-08-11
> **Response to Reviewer 7wHj**
>
> We thank the reviewer for the valuable comments and suggestions.
>
> >"A discussion on the tradeoffs of setting alpha close to 1 or away from 1 will be useful."
>
> The impact of the choice of $\alpha$ can be observed from the results in Theorem 1 and 2. In general, when $\alpha$ is approaches 1 the algorithm allows only one cycle per batch, which increases the number of batches and leads to better regret bounds. Note that as $\alpha\rightarrow 1$, the bounds on batch complexity in (7) and (8) explode and the only remaining guarantee on the number of batches is the problem dependent bound in (9). This essentially implies that the batch complexity can be large for some problem instances (when $\Delta_i$ is small). On the other hand, large $\alpha$ results in better guarantees on batch complexity in (7) and (8) but leads to larger regret bounds in Theorem 1.
>
>
> >"A discussion on how the regret analysis with K=2 will be extended to K > 2 in the full proof will be insightful. With the cycle structure, the transition from K=2 to K > 2 seems non-trivial."
>
> As the reviewer correctly pointed out, the transition from $K=2$ to $K>2$ is non trivial. As we have shown in the main paper, the regret analysis for the $K=2$ case simplifies due to the simple structure of each cycle in that case: in each cycle we sample both the optimal and the suboptimal action once. The fact that both the optimal and  the suboptimal arm are sampled the same (deterministic) number of times naturally leads to easier tail bounds and allows the algorithm to estimate their mean with the same confidence. In the $K>2$ case however, the number of times each arm is sampled is random, which complicates the analysis. We deal with this by decomposing the regret as in eqs (74) to (79). We will  discuss the main ingredients of this extension in the revision.
>
> >"The experiments are not run for long enough time, as evident from the linear regret shape. I suggest running the algorithm for enough time to obtain the log(T) shaped curve and use logscale for Y-axis."
>
> We will run the experiments for a longer duration as suggested by the reviewer and add couple of other metrics and simulations suggested by other reviewers.
>
> >"In Theorem 1 and Theorem 2 please specify that $\alpha$ > 1."
>
> Thanks for pointing this out. We will correct it in the revision.

---

### Official Review · Reviewer_jTWC · 2021-07-01

**Rating:** 6
**Confidence:** 4

**Summary:**

The paper tackles the batched multi-armed bandit problem, where arms are played in batches and feedback is obtained by the end of each batch. The authors suggest the Batched Thompson-Sampling algorithm, an anytime TS algorithm with Gaussian priors that are updated at the end of adaptive batches. The adaptive mechanism for the batch sizes is based on a doubling trick – a batch ends once the number of switches from/to some arm is doubled (or increases geometrically). The authors prove that the batched feedback degrades the regret only by a constant factor and show that the number of batches is logarithmic a.s. and is $O(\log\log T)$ in expectation, for fixed gaps and large enough T.
The authors claim that this work is the first to achieve tight problem-dependent and independent regret bounds with a batch number that adapts to the specific instance ($\log T$ problem independent and $\log\log T$ problem-dependent batch number).

**Ethical Concerns:**

None.

**Limitations And Societal Impact:**

In my opinion, the authors did not adequately discuss the limitations in this work – see all previous comments on the guarantees on the batch number.

**Main Review:**

Overall, I think that the paper has two interesting contributions. First, the cycle (arm-switch)-based doubling trick, which might be extended to other algorithms for adaptive batch-sizing. Second, the application of TS to the batched MAB problem. However, there are a few issues with the results that the authors did not discuss.

One main issue is the guarantees on the number of batches. First, in previous works, the $\log T$ #batches regime was required for the problem-dependent regret bounds while $\log\log T$ batches were required for the problem-independent regret bounds, which are mainly relevant for smaller gaps. However, in this work, the $\log\log T$ batch-bounds are achieved only when the gaps are large. For example, when $\Delta\approx1/\sqrt{T}$ (which is the interesting regime for problem-independent bounds), the batch number is of order $\log T$ and not $\log\log T$, as in previous works. In a sense, this means that the result is not a ‘best of both worlds’ result, as might be implied from its description, but rather a compromise in the problem-independent regime and (possibly) an improvement in the problem-dependent regime.

A natural follow-up issue regards the relation between the upper and lower bounds for the number of batches, a comparison that was not done in this paper. In [10], the authors proved $\log T$ and $\log\log T$ lower bounds for the batch number in the problem-dependent and independent regimes, respectively. Seemingly, the problem independent performance is loose, while the problem-dependent one violates the lower bounds. This isn’t necessarily the case, as the gap-dependence in the lower bound is not entirely clear, but I expect the authors to discuss and justify it. Similarly, the authors should emphasize that some of the guarantees in this paper are weaker than ones in previous work, as they only hold in expectation and not a.s. (e.g., when comparing the a.s. $\log\log T$ guarantees). Namely, the gap dependence comes at the price of a weaker type of guarantee.

Finally, the authors assumed that the optimal arm is unique. In many cases of TS proofs, this is a technical assumption that simplifies analysis (and coupling arguments allow removing it, e.g., Agrawal and Goyal, 2012). However, in this work, it might completely break the batch division mechanism, as there would be many short cycles with two optimal arms. In particular, I believe that both expected bounds on the number of batches would not hold in this case. If this is indeed the case, it should be clearly reflected and justified in the paper, and in my opinion, it greatly weakens the results.

Other comments:
- I didn’t find the explanation on the usage of Gaussian priors, in contrast to Beta priors, very convincing (is there no gain in using KL-UCB vs. UCB1?) - is there a better reason (possibly a technical one)?
- Simulations: an interesting missing statistic is the maximal number of batches in each scenario. Moreover, I think the evaluated algorithm should be the one in which the bounds hold.
- In my opinion, it would be beneficial to add a summary + discussion (e.g., on all previously mentioned issues) by the end of the paper (for example, instead of the simulations of the Gaussian case, that I think that can safely move to the appendix).

Minor comments:
- Line 62 – minimax optimal up to log-factors
- Line 120 – an batching -> a batching
- Line 143 – min -> inf
- Looking at the proof of (7), isn’t there a (+1) missing inside the log?
- Lines 218-220 – add commas.
- Line 256 – the second -> the second arm
- Checklist I -> we


References:

Shipra Agrawal and Navin Goyal. 2012. Analysis of Thompson sampling for the multi-armed bandit problem. In Proceedings of the 25th Annual Conference on Learning Theory (COLT’12)

#########

I thank the authors for the response. I decided to leave my score unchanged.


**Time Spent Reviewing:**

8

---

> ### Author Response · Authors · 2021-08-11
> **Response to Reviewer jTWC**
>
> We thank the reviewer for the valuable comments and suggestions.
>
> >"The result is not a ‘best of both worlds’ result, as might be implied from its description, but rather a compromise in the problem-independent regime and (possibly) an improvement in the problem-dependent regime."
>
> We note that here we are mainly interested in anytime batched algorithms which do not require the knowledge of the time horizon T to design the batch structure (and tune it to the specific regret bound).  As the reviewer said the previous works show that one can achieve the optimal minimax rate with only $\log\log(T)$ batch complexity. However, those batched algorithms require the knowledge of the time horizon T to optimize both batch sizes and the confidence intervals. While the doubling trick in [5] can be used to convert these algorithms into anytime algorithms,  this comes at the expense of increasing the batch complexity from $\log\log(T)$ to $\log T$. We discuss this in the last paragraph of our introduction. In that sense, we believe our result is a ‘best of both worlds’ result as it achieves the batch complexity achieved by other anytime batched strategies (or strategies converted to an anytime strategy with the help of the doubling trick)  and at the same time presents an improvement in the problem dependent regime.
>
>
>
> >"In [10], the authors proved $\log(T)$ and $\log\log(T)$ lower bounds for the batch number in the problem-dependent and independent regimes, respectively. Seemingly, the problem independent performance is loose, while the problem-dependent one violates the lower bounds."
>
>  We note that the $\log\log(T)$ lower bound on the batch complexity in the problem independent regime was proven for strategies that take the time horizon $T$ as an input. We believe  that there is a gap between the batch complexity of anytime algorithms and algorithms that optimize the regret for a given time-horizon T. In other words, we believe that the $\log\log(T)$ lower bound in the problem independent regime cannot be achieved by anytime strategies. However, proving improved lower bounds for the problem-independent batch complexity (possibly of order $\log T$) of anytime algorithms currently remains an open problem in the case of the minimax analysis.
>  As for the problem-dependent performance, there is no inconsistency between the $\log(T)$ lower bound from earlier works and our expected $\log\log(T)$ batch complexity guarantee. This is because the problem dependent regret definition in [10] takes a supremum over all possible reward distributions for a given $T$. This suggests that the "worst case" reward distribution may be different for various $T$ values. On the other hand, our expected regret bound holds  for all $T$ values and for a given fixed reward distribution, i.e. the reward gaps.
>
> >"The authors should emphasize that some of the guarantees in this paper are weaker than ones in previous work, as they only hold in expectation and not a.s. (e.g., when comparing the a.s. $\log\log(T)$ guarantees). Namely, the gap dependence comes at the price of a weaker type of guarantee."
>
> It is true that for a fixed $T$, there are minimax optimal policies with a batch complexity of  $\log\log(T)$ a.s., however as mentioned above their anytime counterparts have a batch complexity of $\log(T)$ a.s. This same guarantee holds for our algorithm in addition to the bounds on the expected batch complexity of our algorithm.
>
>
>
> >"Finally, the authors assumed that the optimal arm is unique. In many cases of TS proofs, this is a technical assumption that simplifies analysis. However, in this work, it might completely break the batch division mechanism, as there would be many short cycles with two optimal arms."
>
> We agree that when the optimal arm is not unique there will be many short cycles and the problem dependent batch complexity guarantees will not hold in this case, however we note that the problem-independent bound in (7) will continue to hold as well as the regret bounds in Theorem 1.
>
>
>
>
>
> >"I didn’t find the explanation on the usage of Gaussian priors, in contrast to Beta priors, very convincing (is there no gain in using KL-UCB vs. UCB1?) - is there a better reason (possibly a technical one)?"
>
> The use of Gaussian priors is more of a technical choice, as it allows us to increase the range of $\alpha$.
>
> >"Simulations: an interesting missing statistic is the maximal number of batches in each scenario. Moreover, I think the evaluated algorithm should be the one in which the bounds hold."
>
> We will add the suggested maximal number of batches in each scenario as well as the original algorithm to our simulations.
>
> >"summary+discussion":
>
> Thank you for this suggestion. We will incorporate it in our revised paper.

---

### Official Review · Reviewer_HuxN · 2021-07-02

**Rating:** 6
**Confidence:** 3

**Summary:**

This paper studies the Thompson Sampling policy for the batched multi-arm bandit problem. The policy only collects the reward at the end of each batch, while the endpoints of each batch are at the discretion of the policy. Both the adaptive and the minimax regret bounds are given, and the bound on the number of batches is also given. The proved bounds are comparable with the bounds of the successive elimination algorithms for batched multi-arm bandits, which shows Thompson Sampling is equally comparable for the batched bandit problems.

**Limitations And Societal Impact:**

The authors compare their algorithm with previous ones and show the advantages over others.

**Main Review:**

Given the batched bandit problem and the batched successive elimination policy in previous works, it is a direct association to consider how well Thompson Sampling performs under the batched setting. This paper answers this question by proving that the batched Thompson Sampling can achieve comparable regret bound as its instantaneous counterpart. The bounds are also comparable with the successive elimination policy in previous works.

The paper is clearly written and easy to read, while the proof of Theorem 1 in the supplementary is a bit difficult for me to follow. It would be helpful to provide a bit more explanation on the decomposition from Equation (74) to (79).

The technical originality is not well highlighted. Given the previous works on batched bandit problem and the fact that Thompson Sampling is well studied before, the relation between this paper and the previous ones is not very clear. This might lead to an impression that this is rigorous work yet without much technical novelty.  For example,  the analysis seems to heavily rely on the geometrical grid, which is also used in previous works like [10] Gao et al. How original is the proof technique used here given the previous work?  Another concern which I am not quite sure about is that the proposed algorithm follows [2] Agrawal and Goyal. What is the relation between this paper's analysis and [2]'s. Some clarification in terms of technical contributions aside from the algorithmic design and performance advantage would be helpful.

**Time Spent Reviewing:**

6

---

> ### Author Response · Authors · 2021-08-11
> **Response to Reviewer HuxN**
>
> We thank the reviewer for the valuable comments and suggestions.
>
> >"How original is the proof technique used here given the previous works [with respect to [10] and [2]]?"
>
> Our technical analysis differs from the earlier work in three important ways:
>
>
>    1) In section D, we provide tail bounds in the case of a random number of observations for each arm. Note that a major complication in our algorithm is the fact that we do not utilize the rewards corresponding to all selected actions by the TS algorithm. As a result the number of reward samples observed for each arm is random which makes it difficult to provide tail bounds for each arm. This is unique to our algorithm because  earlier works such as BaSE [10] and [2] handle the probability guarantees either with the help of the round robin action selection in [10], which results in the same (deterministic) number of observations for each arm, or by the fact that each time an action is selected its reward is observed as in the classical TS setting [2].  We prove our tail bounds using the classical martingale analysis with random stopping times and the fact that the number of observations can not increase more than $\alpha$ from one batch to the next, which allows us to give a deterministic range on the number of samples.
> 2) We introduce a novel batch complexity analysis that is centered around the definition of cycles (See the proof of Theorem 2). Using the fact that each arm has to be accompanied by another in each cycle, we show that bounding the regret naturally leads to a bound on the batch complexity. As far as we know, this approach to bounding the batch complexity is unique to our paper.
> 3) Similarly, the regret analysis also revolves around the cycle mechanism which uses a random stopping time. While random stopping times have been considered before in the context of batched bandits [9,13], these  algorithms employ one or two stopping times while we deal with infinitely many stopping times, i.e. cycle end points. Our analysis is closer to the work of Agrawal and Goyal [2], as we use a modified version of one of the results in their paper, which is Lemma 2.8. However, the main technical novelty arises in how we decompose our regret in the presence of infinitely many stopping times in the equations from (74) to (79). We will discuss this in more detail in the revised version.

---

### Official Review · Reviewer_WhRh · 2021-07-06

**Rating:** 6
**Confidence:** 2

**Summary:**

This paper studies the classical stochastic multi-armed bandits with batches. While the classical problem assumes that rewards are received after each decision step, here the authors consider batches: the decision maker takes a batch of decisions and then observes the rewards at the end of the batch. The next batch of decisions is then taken based on this feedback.

The authors derive an algorithm that uses a limited number of batches ($O(\log(T))$ of $O(\log\log T)$ for a problem-independent or dependent bound) while providing an order-optimal regret ($O(\sqrt{T\log(T)})$ or $O(\log(T))$ for problem-independent or dependent bound). The algorithm is based on Thompson sampling and uses a careful definition of batches.


**Ethical Concerns:**

NA.

**Limitations And Societal Impact:**

The complex definition of the batches and the potential difficulty in implementing it might be worth discussing.

**Main Review:**

I liked the setting and the results presented in the paper. The motivation behind batched-feedback is clear and well explained in the paper. The references to related work seem good. I did not look at the detailed proofs but the approach seems good.

I am quite impressed (and also a bit puzzled) by how the batches are constructed. This seems to be the main difficulty of the paper. If I understand correctly, the authors define batches so that they are bigger when an arm is well identified as being optimal. When multiple arms are considered equivalent by the algorithms, the cycles will be shorter which means that batches will also be smaller. What bothers me a bit in this paper is the complexity of the approach: instead of using Thompson sampling and the very complex definition of cycles, would not it be simpler to use confidence intervals to define a deterministic batch duration?

Also, the algorithm uses a sort of internal feedback because the definition of a cycle uses the sequence of actions taken. This is not a problem in a centralized setting because only the decision taken by the algorithms are requested but not the observed reward. Yet, in a distributed setting (used as a motivation in the paper), this might be problematic. Here again, a simpler definition of batch lengths might be useful.

Last, the authors present two versions of their algorithm: (1) with feedback only at the end of each cycle, or (2) with feedback at every time-step. The authors analyze (1) in theory but present experimental results only for (2). I imagine that the variant (1) performs much worse in practice. This could be studied: how much worse? How many samples are "saved" by doing so?


**Time Spent Reviewing:**

2

---

> ### Author Response · Authors · 2021-08-11
> **Response to Reviewer WhRh**
>
> We thank the reviewer for the valuable comments and suggestions.
>
> >"Instead of using Thompson sampling and the very complex definition of cycles, would not it be simpler to use confidence intervals to define a deterministic batch duration?"
>
> It might be possible to replace the current definition of cycles with an alternative batching scheme based on confidence intervals, however we currently believe that the alternative form will have to essentially implement the role of cycles and will therefore amount to an equivalent batching scheme.  Introducing the notion of confidence intervals is also likely to require additional parameter tuning (of the confidence intervals) while the current definition of cycles is derived from the action selection process of TS. However clearly the possibility of a simpler batching scheme can not be preemptively ruled out and we believe that is an interesting direction for future work. Regarding not using Thompson sampling, while Thompson sampling is our main focus in this paper we believe the idea of cycles as a way to use agent's actions to adaptively increase the batch size, can be combined with other bandit schemes.
>
> >"Implementation of batched TS in a distributed setting might be problematic."
>
> Our batched TS algorithm can indeed be implemented in a distributed setting in the following sense, which we believe covers the applications we mention: the actions to be taken by the TS agent in the next batch can be generated (in an offline fashion) at the beginning of the batch at a central node, and then distributed (even reordered if desired) and applied separately at different nodes. As pointed out by the reviewer this is possible because the batching strategy does not require to know the observed rewards to decide on the end point of the batch and the sampling distribution is only updated at the end of the batch based on the rewards received. In the medical application this corresponds to having the medical expert design her next trial (planned interventions in the next batch) after observing the results of the previous trial and applying these interventions to the set of patients in parallel.
>
>
> >"How much worse does (1) perform compared with (2)?"
>
> Although we do not have an exact theoretical comparison, from the experimental results we can say that the performance relations is of the following order: Regret(2)<Regret(1)<2*Regret(2). We will add experimental results for (1) in the revised version.
>
> >"How many samples are saved by implementing (1) instead of (2)?"
>
> Thanks for bringing up this question. We can prove that the expected number of samples, i.e. number of actions whose rewards are incorporated by  algorithm (1), is upper bounded by $O(\min(\sum_{i=2}^{K}\frac{\log(T)}{\Delta_i^2}, T))$ as follows. We know that there can be at most $N_i(T)$ many samples from each sub-optimal arm, and since the optimal arm is sampled at most once in each cycle where a suboptimal arm has been played,  the number of samples corresponding to the optimal arm is  at most $\sum_{i=2}^K N_i(T)$. Combining these observations with (2) of Theorem 1 shows that the expected number of samples used by (1) is of order $O(\min(\sum_{i=2}^{K}\frac{\log(T)}{\Delta_i^2}, T))$ while (2) uses rewards of all actions so the number of samples corresponding to (2) is clearly $T$. This result shows that (1) uses significantly less amount of samples than (2) if the  gaps are much bigger than $1/\sqrt{T}$.

---

> > ### Comment · Reviewer_WhRh · 2021-08-31
> > **Post rebuttal**
> >
> > I would like to thank the authors for their clear answer. I half-convinced by the discussion about the distributed setting but the others answers are clear.

---

### Decision · Program_Chairs · 2021-09-27

**Decision:**

Accept (Poster)

**Comment:**

In this paper, the authors propose an interesting batch Thomson Sampling algorithm that achieve the optimal regret with $O(\log T)$ and $O(\log\log T)$ number of batches in instance dependent and instance independent settings. I very much like the results and propose acceptance.

I would like to bring the authors' attention to two concurrent work on the exact same topic:

"Parallelizing Thompson Sampling", https://arxiv.org/abs/2106.01420
"Batched Thompson Sampling for Multi-Armed Bandits", https://arxiv.org/abs/2108.06812

I propose the authors discuss the abovementioned work in the final version of their paper to highlight the differences in algorithm design and the corresponding results. I believe it adds to the value of their work.